# A comprehensive non-redundant gene catalog reveals extensive within-community intraspecies diversity in the human vagina

Bing Ma[1,2], Michael T. France[1,2], Jonathan Crabtree [1], Johanna B. Holm[1,2], Michael S. Humphrys[1], Rebecca M. Brotman [1,3] & Jacques Ravel [1,2✉]

Analysis of metagenomic and metatranscriptomic data is complicated and typically requires extensive computational resources. Leveraging a curated reference database of genes encoded by members of the target microbiome can make these analyses more tractable. In this study, we assemble a comprehensive human vaginal non-redundant gene catalog (VIRGO) that includes 0.95 million non-redundant genes. The gene catalog is functionally and taxonomically annotated. We also construct a vaginal orthologous groups (VOG) from VIRGO. The gene-centric design of VIRGO and VOG provides an easily accessible tool to comprehensively characterize the structure and function of vaginal metagenome and meta-transcriptome datasets. To highlight the utility of VIRGO, we analyze 1,507 additional vaginal metagenomes, and identify a high degree of intraspecies diversity within and across vaginal microbiota. VIRGO offers a convenient reference database and toolkit that will facilitate a more in-depth understanding of the role of vaginal microorganisms in women's health and reproductive outcomes.

[1] Institute for Genome Sciences, University of Maryland School of Medicine, Baltimore, MD 21201, USA. [2] Department of Microbiology and Immunology, University of Maryland School of Medicine, Baltimore, MD 21201, USA. [3] Department of Epidemiology and Public Health, University of Maryland School of Medicine, Baltimore, MD 21201, USA. ✉email: jravel@som.umaryland.edu

The microbial communities that inhabit the human body play critical roles in the maintenance of health, and dysfunction of these communities is often associated with disease[1]. Taxonomic profiling of the human microbiome via 16S rRNA gene amplicon sequencing has provided critical insight into the potential role of the microbiota in a wide array of common diseases[2,3], yet these data routinely fall short of describing their etiology, including bacterial vaginosis[4], Crohn's disease[5], or psoriasis[6], among others. This is perhaps because while 16S rRNA gene sequencing can provide species-level taxonomic profiles of a community, it does not describe the genes or metabolic functions that are encoded in the constituents' genomes. This is an important distinction because strains of a bacterial species often exhibit substantial diversity in gene content[7], such that their genomes harbor sets of accessory genes whose presence is variable[8]. It is therefore difficult, if not impossible, to infer the complete function of a microbial species in an environment using only the sequence of their 16S rRNA gene. As a consequence, to investigate the role of the human microbiome in health and diseases, particular emphasis should be placed on describing the gene content and expression of these microbial communities.

Metagenomic and metatranscriptomic profiling are emerging approaches aimed at characterizing the gene content and expression of microbial communities. Results have led to increased appreciation for the important role microbial communities play in human health and diseases[9,10]. Despite the rapid development and increased throughput of sequencing technologies, current knowledge of the genetic and functional diversity of microbial community is still limited. This is due, at least in part, to a lack of resources necessary for the analysis of these massive datasets[11]. De novo assembly of metagenomic or metatranscriptomic datasets typically requires rather substantial computational resources and complicates integration of metagenomic and metatranscriptomic data.

Accurate, high-resolution mapping of metagenomic or metatranscriptomic data against a comprehensive and curated gene database is an alternative analytical strategy that is less computationally demanding, prone to fewer errors, and provides a standard point of reference for comparison of these data. Development of such curated databases is crucial to further our understanding of the structure and function of microbial communities[12]. In the last two decades, international initiatives such as MetaHit, the NIH funded Human Microbiome Project (HMP) and the International Human Microbiome Consortium (IHMC) were established to generate the resources necessary to enable investigations of the human microbiome, including large reference taxonomic surveys and metagenomic datasets[9]. While multiple 16S rRNA gene catalogs exist[13,14], there are relatively few curated resources for referencing metagenomes and metatranscriptomes. Those that do exist focus only on the gut microbiome of either humans[12,15] or animal model species[16]. A definite unmet demand exists for reference gene catalogs for other body sites such as the oral cavity, the skin, and the vagina[17].

In this study, we constructed the human vaginal non-redundant gene catalog (VIRGO), an integrated and comprehensive resource to establish taxonomic and functional profiling of vaginal microbiomes from metagenomic and metatranscriptomic datasets. VIRGO was constructed using a combination of metagenomes and urogenital bacterial isolate genomes. The genes identified in these data were further clustered into Vaginal Orthologous Groups (VOGs), providing a catalog of functional protein families common to vaginal microbiomes. We meticulously curated the gene catalog with taxonomic assignments as well as functional features using 17 diverse protein databases. Importantly, we show that VIRGO provides >95% coverage of the human vaginal microbiome, and it is applicable to populations from North America, Africa, and Asia. Together, VIRGO and VOG represent a comprehensive reference repository and a convenient cataloging tool for fast and accurate characterization of vaginal metagenomes and metatranscriptomes. The gene catalog is a compilation of vaginal bacterial species pan-genomes, creating a vaginal meta-pan-genome. We further used VIRGO to characterize the amount of intraspecies diversity present in individual vaginal communities. Previous characterization of these communities using either 16S rRNA gene taxonomic profiling or assembly based metagenomic analyses has failed to resolve this diversity. Here we show that vaginal communities contain far more intraspecies diversity than originally expected. This observation challenges the notion that the vaginal microbiota dominated is by one species of *Lactobacillus*, comprised of a single strain, and could have major implications for the ecology of these otherwise low-diversity bacterial communities. Ultimately, VIRGO and its associated analytical framework will facilitate and standardize the analysis and interpretation of large metagenomic and metatranscriptomic datasets thus expanding our understanding of the role of vaginal microbial communities in health and disease.

## Results

**VIRGO is sourced from comprehensive vaginal sequence datasets**. VIRGO was constructed using sequence data from fully de-identified vaginal metagenomes ($n = 264$) as well as complete and draft genomes of urogenital bacterial isolates ($n = 308$). The majority ($n = 211$) of the included metagenomes were sequenced in-house from de-identified vaginal swab specimens. Of the ~18 billion reads generated for these metagenomes, 14.4 billion (79.7%) were identified as human sequences and removed. We found that vaginal metagenomes dominated by *Lactobacillus* spp. had significantly higher proportions of human sequence reads than those from *Lactobacillus* deficient metagenomes (88.7% vs. 73.3%; $t = -6.6$, $P < 0.001$; Supplementary Fig. 1). Each metagenome was then de novo assembled totaling 1.2 million contigs with a combined length of 2.8 billion bp and an N50 of 6.2 kbp. Additional metagenomic data ($n = 53$) were obtained from the HMP[9] and contributed 40,000 contigs, comprising 100 million bp of assembled sequence. The in-house metagenomes provided 19.5 times more assembled length than the HMP vaginal metagenomes. In addition to the vaginal metagenomes, we also included 308 complete or draft genome sequences of urogenital bacterial isolates obtained from GenBank and IMG/M (Integrated Microbial Genomes & Microbiomes)[18]. A summary of the metagenomic reads, assembled contigs and genomes included in the construction of VIRGO can be found in Supplementary Data 1).

Taxonomic analysis of the metagenomes included in VIRGO using MetaPhlAn (v2)[19], revealed that these communities contained 312 bacterial species present in ≥0.01% relative abundance (Supplementary Data 2). All major vaginal *Lactobacillus* species (*L. crispatus*, *L. gasseri*, *L. iners*, and *L. jensenii*), as well as common facultative and strict anaerobic vaginal species such as *Gardnerella vaginalis*, *Atopobium vaginae*, *Prevotella amnii*, *Megasphaera* genomosp., *Mobiluncus mulieris*, *Mageebacillus indolicus* (aka. BVAB3), *Veillonella parvula*, among others were identified in the metagenomes. Even BV-associated bacteria that are often only present at low abundance[20] were represented, including *Finegoldia magna*, *Peptoniphilus harei*, *Peptostreptococcus anaerobius*, *Mobiluncus curtisii*, *Peptoniphilus lacrimalis*, *Anaerococcus tetradius*, *Ureaplasma urealyticum*, *Veillonella atypica*, and *Corynebacterium glucuronolyticum*. The taxonomic profiles of 264 metagenomes were further shown to encompass the five previously reported vaginal community state types

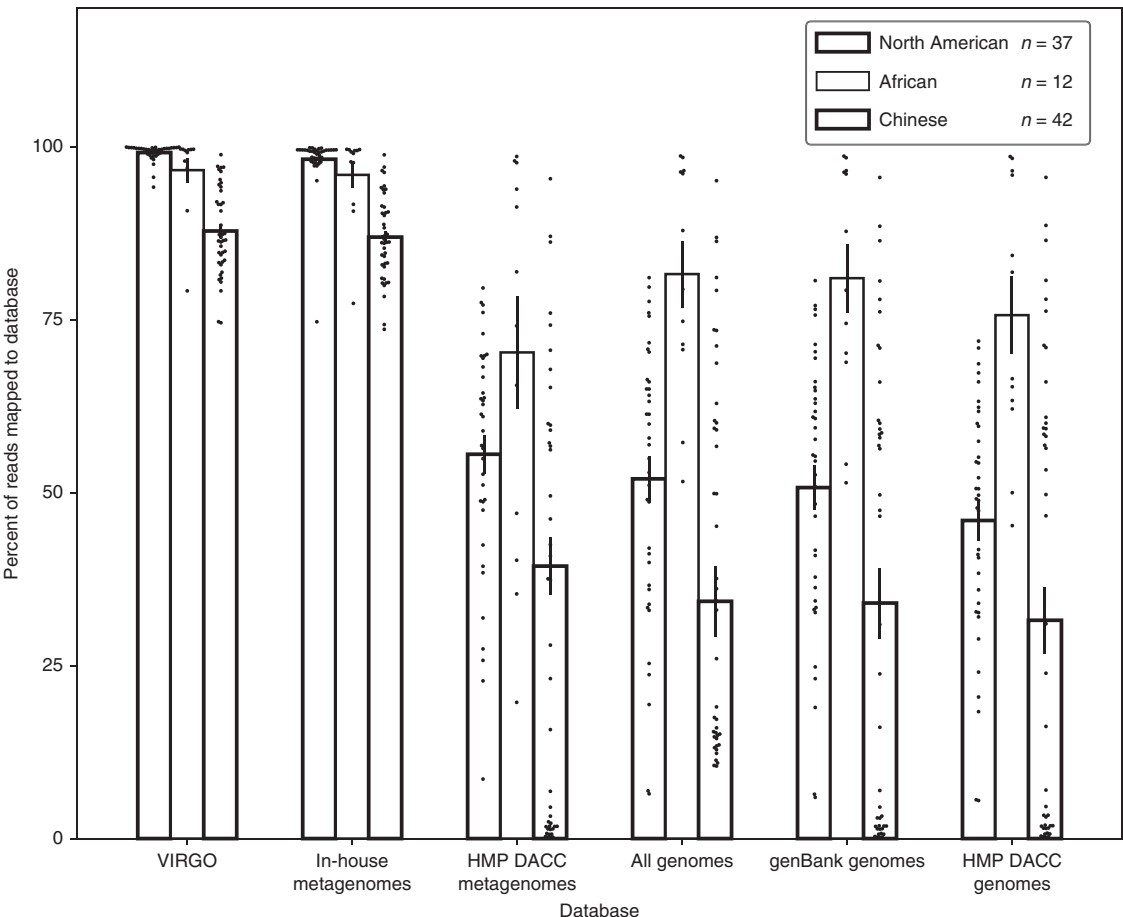

**Fig. 1 Percent of vaginal metagenome reads that can be mapped to contigs from the following reference data sets.** Complete VIRGO database, 211 in-house sequenced vaginal metagenomes, 53 HMP DACC vaginal metagenomes[24], all HMP urogenital reference genomes, 277 genomes of bacteria isolated from vagina, reproductive or urinary system deposited in GenBank, and 139 genomes of urogenital bacteria from HMP DACC database[11]. Values plotted are the average, error bars represent the standard error of the mean.

(CSTs)[21], CST I (*L. crispatus*-dominated), II (*L. gasseri*-dominated), III (*L. iners*-dominated), IV (array of strict and facultative anaerobes), and V (*L. jensenii*-dominated) with frequencies in this set of metagenomes of 18.9%, 3.8%, 20.5%, 48.5%, and 8.3%, respectively (Supplementary Fig. 2. Supplementary Data 2). These results highlight the taxonomic breadth of the vaginal bacterial communities included in the construction of VIRGO (Supplementary Fig. 3).

The dataset used to build VIRGO was compiled from vaginal metagenomes that were obtained from North American women. To determine the comprehensiveness of VIRGO, we mapped reads from 91 vaginal metagenomes that were not included in its construction. These metagenomes were obtained from North American, African[22], and Chinese[23] women, allowing us to determine the utility of VIRGO to analyze metagenomes from other populations. Reads from these metagenomes were mapped to the complete set of sequence contigs used to build VIRGO as well as different subsets of these contigs based on the dataset from which they originated (Fig. 1). More than 99% of the reads from North American metagenomes were able to be mapped to the complete VIRGO dataset, while only ~55% of these reads mapped to contigs from the HMP vaginal metagenomes subset (Fig. 1, Supplementary Data 3). This result indicates a lack of genetic diversity in the HMP vaginal metagenomes, which were derived from highly selected and otherwise healthy women[24]. Further, despite originating from populations not used in the construction of VIRGO, 96 and 88% of the reads from African and Chinese women mapped to the complete VIRGO dataset. For these two cohorts, 71.7% and 99.9% of the reads that failed to map to VIRGO, also did not have a match in GenBank (Supplementary Fig. 4). Interestingly, the African metagenomics mapped onto HMP and isolate genome subsets better than the North American or Chinese datasets (Fig. 1). This may reflect the limited number of reads per metagenome for these samples. These results illustrate the comprehensiveness of VIRGO and its broad application to different populations and ethnicities.

**VIRGO: a non-redundant vaginal bacterial gene catalog.** Coding sequences (CDS, $n = 5,509,298$) were predicted from the metagenomic assemblies and genome sequences using MetageneMark[25]. The core workflow to identify and cluster these CDSs is shown in Fig. 2, and a more detailed illustration is provided in Supplementary Fig. 5. Metagenomic assemblies contributed ~80% of the CDSs while the remaining ~20% originated from urogenital bacteria isolate genome sequences. Redundant genes were then identified and removed via a greedy pairwise comparison at the nucleotide level using highly stringent criteria of 95% identity over 90% of the shorter gene length[12,15]. This process afforded the removal of partial genes and eliminated overcalling genes as unique because of sequencing errors. A total of 948,158 non-redundant CDSs longer than 99 bp were identified and retained, representing 17.2% of the original 5.5 million CDSs. The in-house vaginal metagenomes used to build VIRGO contributed 12 times

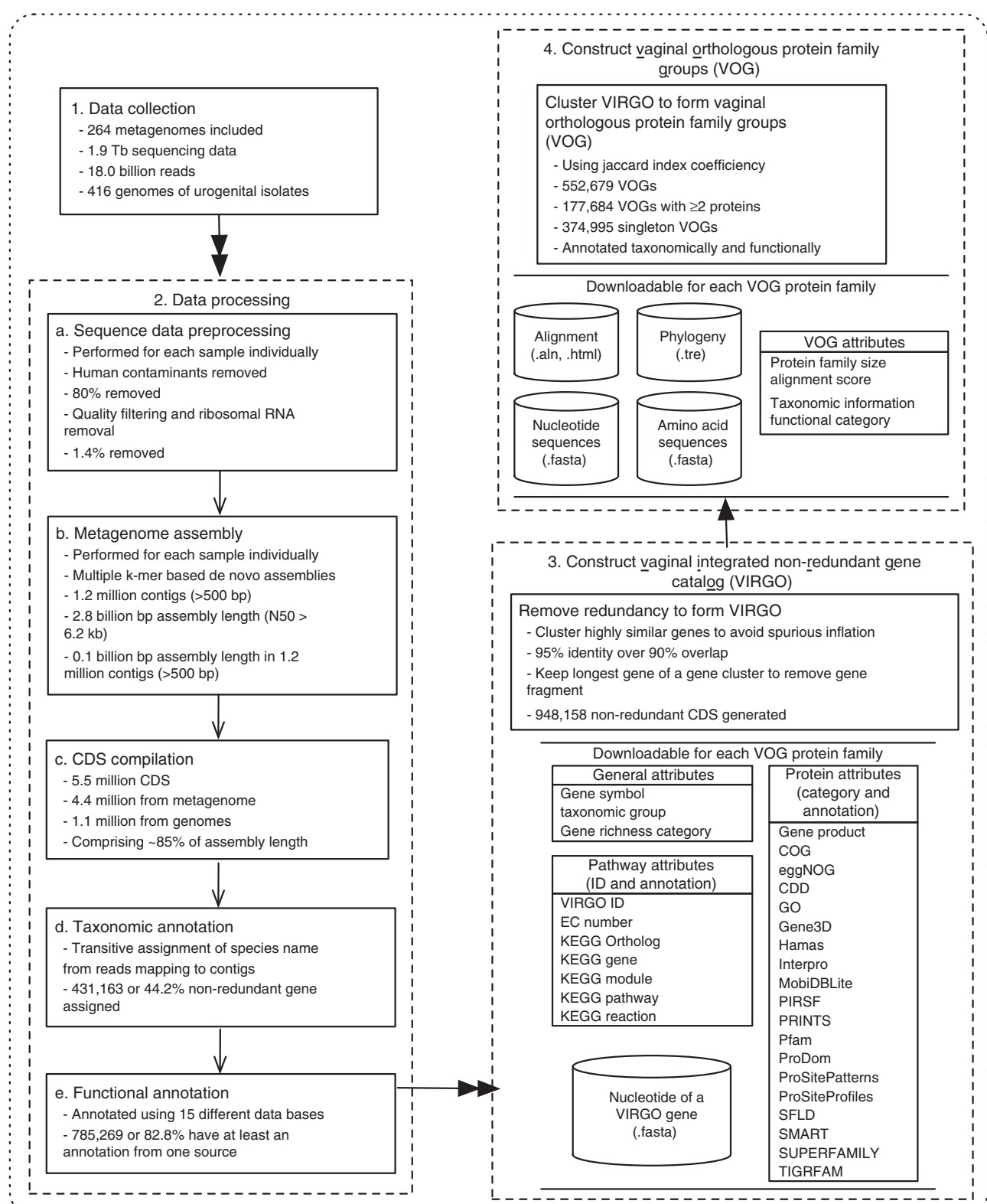

**Fig. 2 Pipeline for data processing and integration for the construction of the human vaginal integrated non-redundant gene catalog (VIRGO) and vaginal orthologous groups (VOG) for protein families.** Metagenomes from 264 vaginal metagenomes and 416 genomes of urogenital isolates were processed, that including 211 in-house sequenced vaginal metagenomes. The procedures include preprocessing to remove human contaminates, quality assessment, metagenome assembly, gene calling, functional, and taxonomic annotation, gene clustering based on nucleotide sequencing similarity to form VIRGO, and jaccard index coefficiency clustering of amino acid sequences to form VOG. A more detailed illustration is in Supplementary Fig. 5 and description is in Material and Methods section.

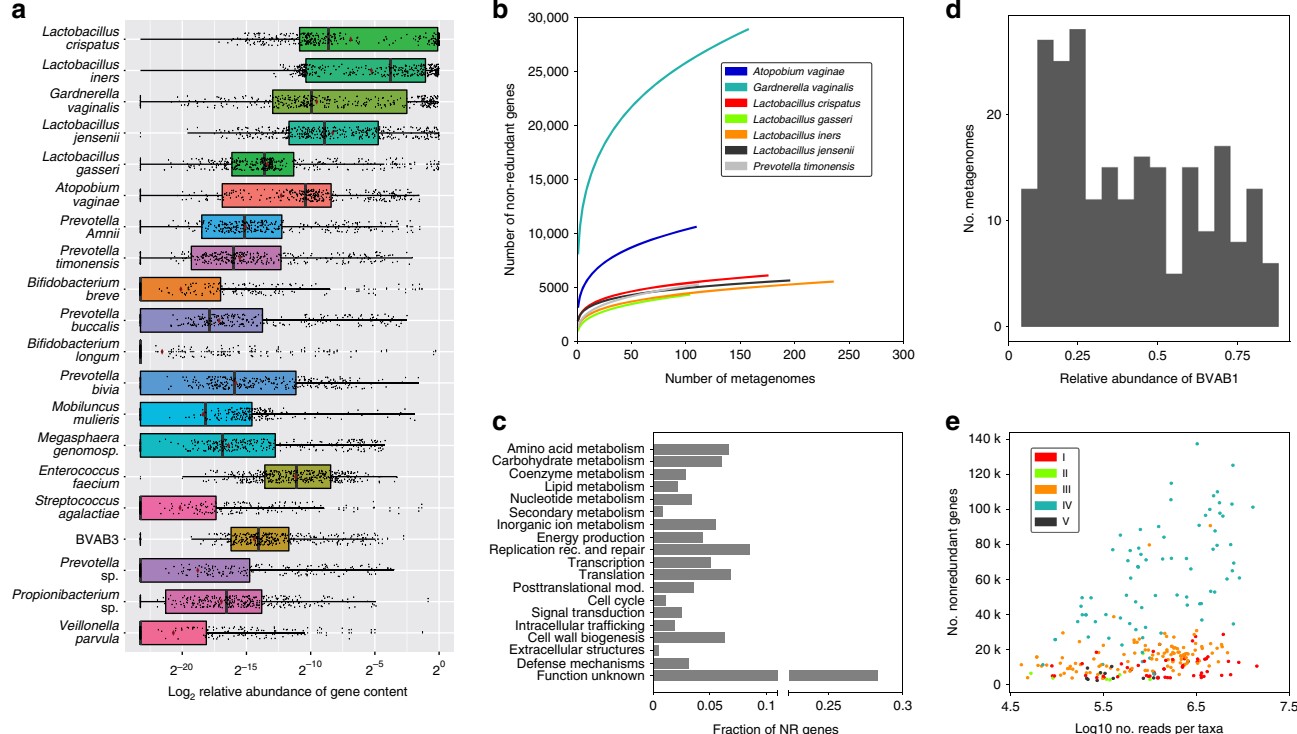

**Fig. 3 Taxonomic and functional composition of vaginal microbiome in VIRGO. a** Top 20 species with the most abundant gene content in VIRGO. The logarithm of the ratio of the gene content of a species over the entire community to the base 2. Plotted are interquartile ranges (IQRs, boxes), medians (line in box), and mean (red diamond). **b** Species-specific metagenome accumulation curves for the number of non-redundant genes. **c** Functional distribution of non-redundant genes in VIRGO. Functional categories were defined using EggNOG (v4.5)[59]. **d** Prevalence of BVAB1 in metagenomes using a minimum number of genes threshold of 50% of the estimated BVAB1 genome size. A gene was present if ≥3 reads mapped to it. **e** Relationship between the depth of sequencing and the number of bacterial non-redundant genes identified using VIRGO. Each point is a separate metagenome and is color-coded according to community state type.

more non-redundant genes (634,288 genes) than the HMP vaginal metagenomes (54,500 genes). Combined, the metagenomes contributed twice as many non-redundant genes as urogenital bacterial isolate genome sequences (371,099 genes). This is likely reflective of the diversity of unculturable bacteria in the metagenomes that is not carried in the isolate genomes.

In order to facilitate the use of VIRGO to characterize vaginal microbial communities, each non-redundant gene was taxonomically and functionally annotated. Non-redundant genes were assigned to taxonomic groups using a custom pipeline as depicted in Supplementary Fig. 5. First, metagenomic contigs were assigned taxonomy if 95% of the composite reads were annotated to the same species. Second, genes encoded on an metagenomic contig with assigned taxonomy were given the taxonomy of that contig (details in Methods section). A total of 445,739 non-redundant genes comprising 47.0% of VIRGO were able to be taxonomically annotated. Overall, 267 unique bacterial species were annotated in VIRGO (Supplementary Data 4), representing a majority of the described vaginal species (Supplementary Fig. 2). This includes BVAB1, an as of yet unculturable vaginal species, for which a closed genome and several metagenome-assembled genomes (MAGs) were recently made available[26–28]. When stratified by CST, CST IV metagenomes have the smallest proportion (<30%) of their gene content taxonomically annotated (Supplementary Fig. 6) compared to ~45–50% in *Lactobacillus*-dominated CSTs. The most abundant species based on gene content are shown in Fig. 3a and Supplementary Fig. 7. Besides bacteria, we also curated potential fungal and phage genes (details in Methods) that were generally present in low abundance if detected at 0.17 ± 0.04% and 0.03 ± 0.001%, respectively. An

additional 10,908 fungal and 15,965 phage genes were included (Supplementary Data 5, https://github.com/ravel-lab/VIRGO).

By including many metagenomes and bacterial isolate genome sequences, we sought to capture each vaginal species' pangenome in VIRGO. To determine the extent to which we were successful, we generated metagenome accumulation curves for the number of non-redundant genes belonging to several key vaginal species (Fig. 3b) These curves track the number of new non-redundant genes added when increasing numbers of metagenomes containing a given species are included in constructing the database. The accumulation curves for six of the seven species tested (*L. crispatus*, *L. iners*, *L. gasseri*, *L. jensenii*, *P. timonensis*, *A. vaginae*) have neared saturation (Fig. 3b). This indicates that VIRGO includes the majority of these species pangenomes. The number of non-redundant genes included for five out of these six species are similar (~5000 genes), while the sixth, *A. vaginae*, had twice as many. This pales in comparison to the number of non-redundant genes included in VIRGO for *G. vaginalis*, which surpasses 25,000 genes. Previous studies have highlighted the remarkable diversity of this species[29–32]. *G. vaginalis* is the only species analyzed which, as estimated by metagenome accumulation curves, did not approach saturation.

The community gene content, or gene richness, can also be characterized by VIRGO as the number of non-redundant genes as recent gut quantitative metagenomics studies[33,34]. As demonstrated in Fig. 4a, *Lactobacillus*-dominated communities were typically categorized as low gene count (LGC) as 82.9% of them have less than 1000 genes (4920 ± 151.6); *Lactobacillus*-deficient communities commonly have high gene count that 88.3% of them have more than 1000 genes (29,898 ± 1025). Further, a vaginal

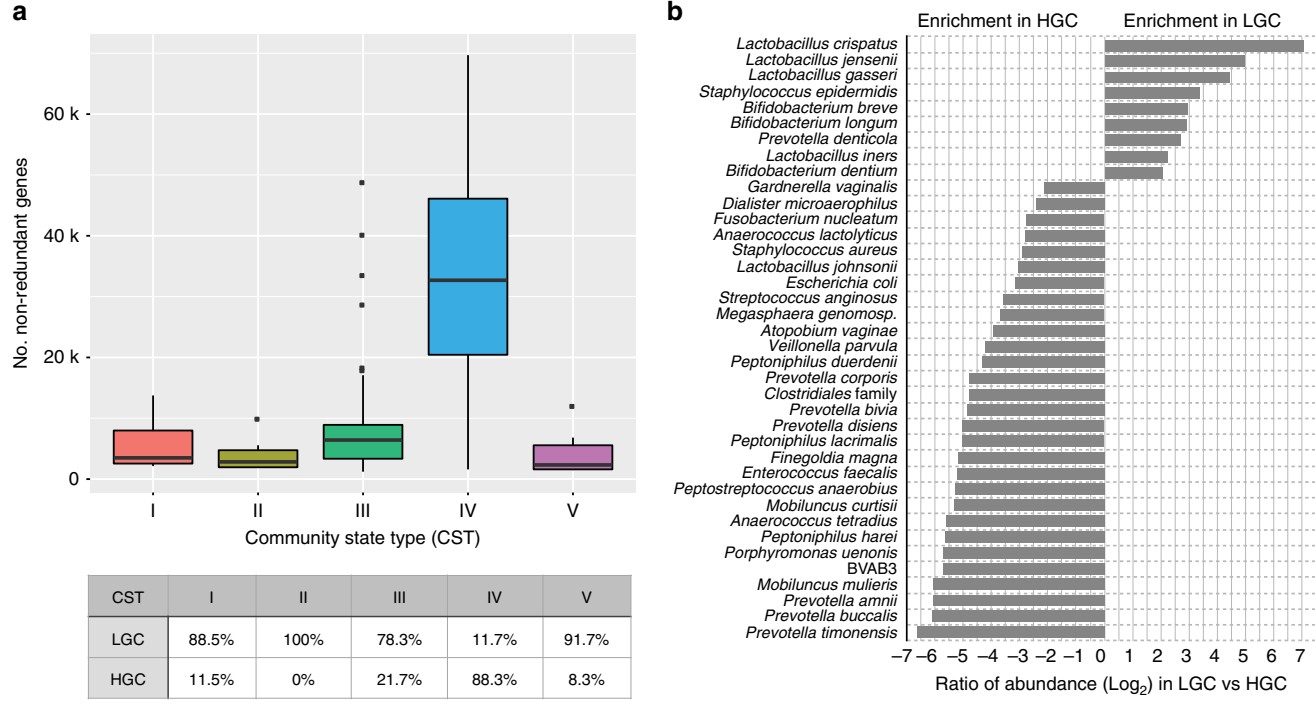

**Fig. 4 a** Boxplot of the number non-redundant genes in samples of different Community State Types (CSTs). Boxes represent the interquartile ranges and lines represent the median values. CSTs were defined as previously according to the composition and structure of the microbial community[21]. Table below boxplot contains percentage of samples in each of the CSTs stratified by high gene count (HGC) or low gene count (LGC), in which HGC has >10,000 non-redundant genes and LGC has <10,000 non-redundant genes. **b** Plot of the $\log_2$ transformed ratio of the gene of a species being in one gene count category over the other across the 264 vaginal metagenomes, only the species with more than four times more abundant in a category (either HGC or LGC) are shown. The species with at least 0.1% abundance and at least 100 genes in either HGC or LGC groups. Plotted are interquartile ranges (IQRs, boxes), medians (line in box), and mean (red diamond).

bacterial species' genes can be overrepresented in HGC or LGC vaginal communities (Fig. 4b) with distinct functional makeup (Supplementary Figs. 10–12, detailed description in Supplementary information file 1 and Supplementary Data 9). *Lactobacillus* spp., particularly *L. crispatus, L. jensenii, L. gasseri, L. vaginalis,* were observed to be highly overrepresented in LGC communities. On the other hand, genes belonging to many other BV-associated species, specifically *P. timonensis, P. buccalis, P. amnii, M. mulieris, Mageeibacillus indolicus, Porphyromonas uenonis, P. harei, Anaerococcus tetradius, M. curtisii,* were overrepresented in HGC communities. Gene richness-based annotations can provide and added dimension to our understanding of the genetic basis of the biological processes that drive vaginal microbiomes.

The non-redundant genes were also annotated with a rich set of functional descriptions. We performed intensive functional annotation using the JCVI standard operating procedure [HMP] for annotating prokaryotic metagenomic shotgun sequencing data[35] as well as 17 additional functional protein databases including KEGG, COG, eggNOG, gene product, CDD, and GO, among others. A complete list of the functional annotation sources employed to characterize the VIRGO non-redundant genes is illustrated in Fig. 2, and an overview of the eggNOG functions encoded in VIRGO is shown in Fig. 3c. Overall 785,268 genes (82.8% of all non-redundant genes) were assigned a functional annotation from at least one source. This gene-rich annotation of the non-redundant gene catalog enables a comprehensive functional characterization of vaginal metagenomes and metatranscriptomes.

**VOG: orthologous protein families in vaginal microbiome.** The non-redundant genes were translated into amino acid sequences

and clustered into vaginal orthologous groups (VOGs). The resulting database of VOGs can be used to investigate the protein families found in the vaginal microbiome. A modified Jaccard index was used as a measure of similarity between amino acid sequences[36]. The similarity between each pair of proteins was calculated as the intersection divided by the union of the list of proteins connected to the pair of proteins, (Fig. 2 and Supplementary Fig. 5 algorithm accessible at https://github.com/ravel-lab/VIRGO). The resulting connected graph of proteins is referred as Jaccard clusters (JACs), and reciprocal best hits of JACs is referred as Jaccard orthologous clusters (JOCs) (details in Methods section). The JOCs orthologous protein families can be highly conserved (alignment score >950) or partially aligned with both conserved and variable regions (alignment score ~300) (Supplementary Fig. 8). This highlights the flexibility of the network-based aggregation algorithm used to recruit both highly similar and distantly related proteins without imposing a single similarity threshold. A total of 617,127 JACs and 552,679 JOCs were generated, of which 177,684 contained at least two genes while the remaining 374,995 are singletons, indicating 38.5% of all VOG proteins are unique.

To demonstrate the utility of VOGs, we retrieved 32 proteins of the orthologous family encoding vaginolysin, a *G. vaginalis* cholesterol-dependent cytolysin that is key to its pathogenicity as it forms pore in epithelial cells[37] (Supplementary Data 6, Supplementary Fig. 9). Using the retrieved alignment, we identified 3 amino acid variants in an 11-amino acid sequences of domain 4 of vaginolysin. One of the three variants, an alanine-to-valine substitution that is divergent across *G. vaginalis* and not been reported previously. This example illustrates how VOG can be mined to understand biological relevance and to generate hypotheses. In this case it points to potential differences in pore

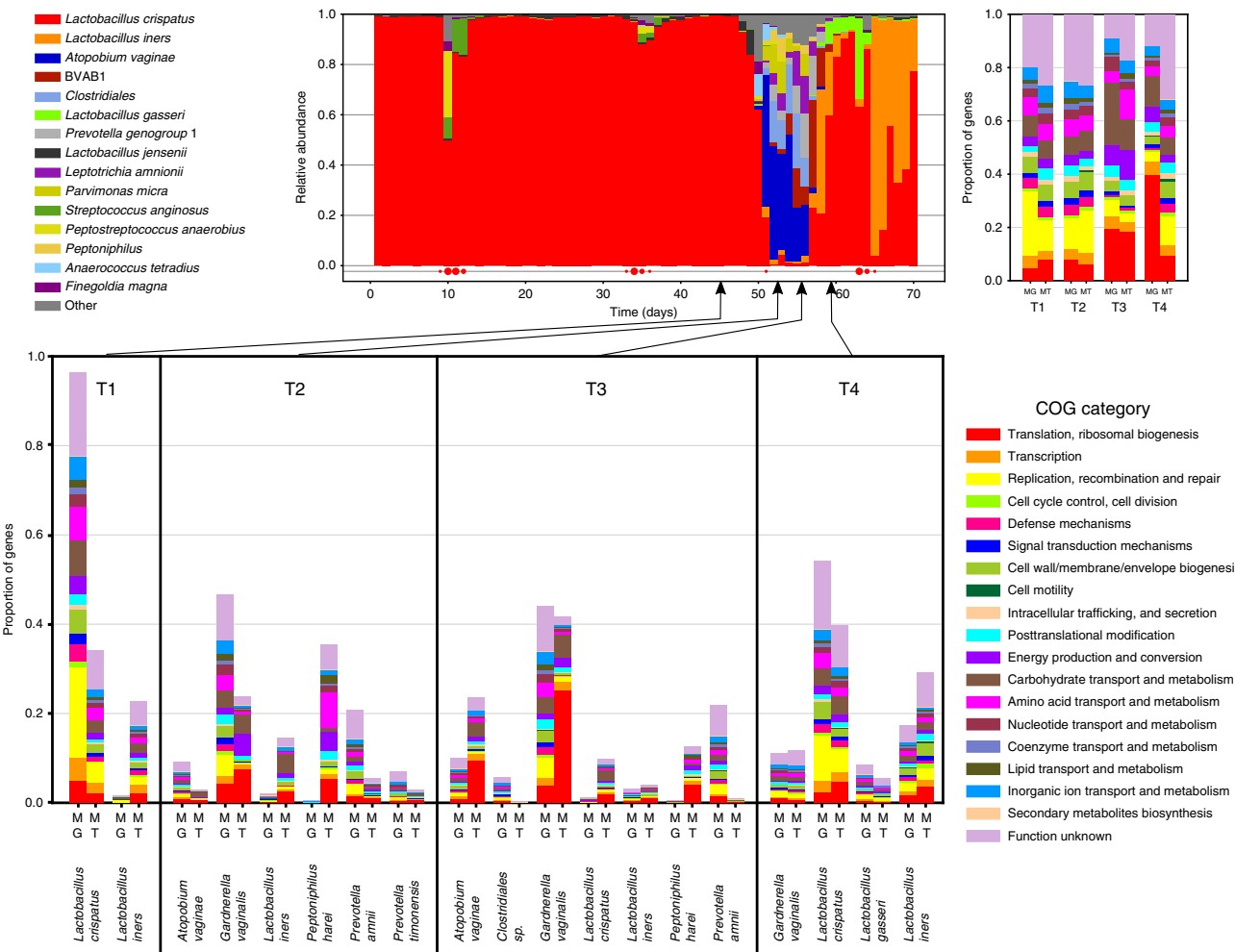

**Fig. 5 Demonstration using VIRGO and VOG to study vaginal microbiome. a** 4 sampling points were selected based on a longitudinally profiled subject prior to (T1), during (T2 and T3), and after (T4) an episode of bacterial vaginosis using 16S rRNA profiling. **b** Functional profiling of the metagenome (MG) and metatranscriptome (MT) of each of the four sampling points. Functional categories were annotated using EggNOG (v4.5)[59]. **c** Functional profiles stratified by species using the taxonomic profiling provided by VIRGO. **d** Demonstrative use of VOG to characterize the *G. vaginalis* cholesterol-dependent cytolysin (CDC) protein family. It shows the phylogeny of CDC-containing protein and alignment of domain 4 of the CDCs that is generally well conserved but contains a single divergent site, highlighted in yellow.

formation activity and possibly cytotoxicity, which could be further investigated. To provide another example of how to use the VOG database to mine genomic information, we searched VOG using the key phrase "cell surface-associated proteins" and "*L. iners*" and retrieved two protein families, one of which was recognized to have an LPXTG motif while the other harbored the motif YSIRK (Supplementary Data 7). Interestingly, a previous study on staphyloccocal proteins suggested that the motifs LPXTG and YSIRK were involved in different biological processes related to surface protein anchoring to cell wall envelope. The two retrieved protein families are specific to *L. iners* and provide relevant evidence for future experimental validation to understand its adherence as previously reported[38]. These two examples demonstrate how the VOG database can be used to explore more mechanistic understandings of vaginal bacterial communities.

**Integration of metagenome and metatranscriptome data using VIRGO.** By serving as a reference, VIRGO enables the characterization and integrative analyses of the abundance of genes and their expression in the vaginal microenvironment. To demonstrate its use, we analyzed a woman's vaginal metagenomes and associated metatranscriptomes at four time points over an

episode of symptomatic bacterial vaginosis (BV): prior to (T1), during (T2 and T3), and after (T4) (Fig. 5a). Not surprisingly, the expressed functions represented in the metatranscriptomes were often different from the encoded functional makeup of the corresponding metagenomes (Fig. 5b). VIRGO also provides rapid binning of genes by species, which revealed dramatic differences in gene abundance and their transcriptional activity in vaginal species (Fig. 5c). Prior to the BV episode (T1), a small proportion of *L. iners* genes were present (1.5%) but these genes exhibited high expression levels, accounting for over 20% of the metatranscriptome. At the same time point, *L. crispatus* genes made up the majority of the gene present (96.3%) but exhibited low expression levels (34.2%). In contrast, at the end of the BV episode (T3), *L. crispatus* gene made up a small proportion of the metagenome but were highly transcriptionally active. This increased activity corresponded with *L. crispatus* regaining dominance at T4, following the resolution of the BV episode. Interestingly, the functions encoded by *G. vaginalis* were similar between T2 and T3 but their expression differed between these timepoints. By enabling this integration of these datatypes, VIRGO can provide a functional understanding of the vaginal microbiota.

**VIRGO revealed high within-community intraspecies diversity**.
VIRGO can be used to characterize the genome content of
individual bacterial species that are present in the vaginal
microbiome. The number of non-redundant genes identified in a
metagenome was not found to correlate with the depth sequen-
cing (Fig. 3e. Supplementary Data 8). We applied VIRGO to a
dataset of 1507 in-house and publicly available vaginal meta-
genomes, to characterize the gene content of four *Lactobacillus*
species (*L. crispatus*, *L. iners*, *L. jensenii*, and *L. gasseri)* and three
additional species commonly found in the vagina (*G. vaginalis*, *A.
vaginae* and *P. timonensis*). We recovered most of each species'
gene content (>80% of the average gene count in a genome) even
when that species was present at low abundance (<1%) in a
community. For instance, even though *P. timonensis*[39] was gen-
erally present in low abundance in these metagenomes (4.8% ±
0.3% mean ± S.E., range [0.1–33.8%]), we recovered the majority
of its genome (2469 ± 401 CDS, Supplementary Fig. 13; Supple-
mentary Data 10). We observed similarly high sensitivity in the
analysis of the other six selected vaginal species (Fig. 6a, Sup-
plementary Data 10). These results demonstrate VIRGO's cap-
ability for characterizing the gene content of low abundance taxa
from metagenomic data.

Using these species-specific gene repertoires, we character-
ized the amount of intraspecies diversity present within an
individual woman's vaginal microbiome. Because VIRGO
comprises the "pangenomes" of each vaginal bacterial species,
it can be used to evaluate the amount of intraspecies diversity
present in these communities. For this analysis, we counted the
number of genes that were assigned to each of the seven species
in each of the 1507 metagenomic datasets and compared this
number to that found in each species' reference genomes. The
number of genes for a species in a community often exceeded
that found in a single isolate genome (Fig. 6a, b), suggesting that
multiple strains of a species co-occur in vaginal bacterial
communities. The total number of *L. crispatus* genes identified
in each of the metagenomes where it was detected contained on
average 1.6 times more genes (3262 ± 586) than that found
encoded on *L. crispatus* genomes (2064 ± 225, $p < 0.001$,
Student's $t$-test). Similar results were observed for *G. vaginalis*,
*A. vaginae*, *L. iners*, *L. jensenii*, and *L. gasseri*. Among these
species, *G. vaginalis* and *A. vaginae* exhibited the highest
within-metagenome intraspecies diversity, while that of *L.
crispatus* was the highest among the major vaginal *Lactobacillus*
spp. (Supplementary Fig. 13 and Fig. 6c). These results suggest
that a woman's vaginal bacterial populations are routinely
comprised of more than one strain of most species. VIRGO
enables the investigation of this unprecedented intraspecies
diversity in vaginal communities.

We next applied well-established practices from pangenomics[7]
in order to identify core and accessory non-redundant genes
among our sample-specific species gene repertoires. Based on the
clustering patterns of gene prevalence profiles, we were able to
define groups of consistently present (core) and variably present
(accessory) non-redundant genes. The majority of the observed
genes for each of the species were categorized as accessory, with
variable representation across the metagenomic datasets. Using *L.
crispatus* as an example, we observed more than twice as many
non-redundant genes with variable representation across the
metagenomes than those present in every sample (Fig. 6c).
Interestingly, it is clear from this analysis that the gene content
identified with VIRGO in genome sequences of *L. crispatus*
under-represent the intraspecies genetic diversity present in the
metagenomes. Similar results were observed for the other six
species analyzed, although the magnitude of the difference
between the metagenome and isolate gene repertoires varied
depending on the species. Overall, VIRGO revealed that

metagenomic data carry a more extensive gene content than is
found in all combined isolate genome sequences.

**Metagenomic subspecies in vaginal ecosystem**. Hierarchical
clustering of the metagenome species-specific gene content pro-
files revealed distinct groupings which we term "metagenomic
subspecies" (MG-subspecies). These metagenomic subspecies
represent types of bacterial populations that share a similar gene
pool as assessed by shotgun metagenomic sequence data. For
example, this analysis revealed at least three distinct metagenomic
subspecies for *L. gasseri* (Fig. 6d). *L. gasseri* MG-subspecies I and
III have large sets of non-redundant genes that are present in one
but not the others, while *L. gasseri* MG-subspecies II carries a
blend of the genes from both MG-subspecies I and III. Our
analysis of *G. vaginalis* revealed more than four MG-subspecies,
concordant with previously studies that had identified multiple
clades within this species[31]. However, we found that the genome-
based paradigm largely under-represents the diversity of *G.
vaginalis* gene content that we identified in metagenomes (Sup-
plementary Fig. 13e). We applied this analysis to seven vaginal
species (Supplementary Fig. 13) and found that vaginal microbial
communities are often composed of complex mixtures of multiple
strains of the same species, and that these mixtures can be clus-
tered into distinct MG-subspecies. Further investigation of these
vaginal MG-subspecies and their gene content is likely to reveal
novel features of vaginal communities and their sub-populations
that will contribute to our understanding of the vaginal ecosystem
of niche-optimized strains.

**Discussion**
Microbiome studies have become increasingly sophisticated with
the rapid advancement of sequencing throughput and the asso-
ciated decrease in sequencing cost. However, identifying features
that drive correlations between the microbiome and health using
multi-omics sequence data remains challenging. This is due, in
part, to difficulties in analyzing and integrating the complex
metagenomic and metatranscriptomic data now common to
microbiome studies. A scalable tool that provides a comprehen-
sive characterization of such multi-omics data is therefore highly
desired. VIRGO is a large vaginal microbiome database designed
to fulfill such research needs for investigations of the vaginal
microbiome and its relation to women's health. In summary,
VIRGO has (i) a comprehensive breadth that includes previously
observed community types, vaginal species, and even fungi and
viruses; (ii) a gene-centric design that enables the integration of
functional and taxonomic characterization of metagenomic and
metatranscriptomic data originating from the same sample; (iii) a
high scalability and low memory requirement; (iv) a high sensi-
tivity that affords characterization of the gene content of low-
abundance bacteria; (v) an easy to use framework from which to
evaluate gene richness and within-species diversity.

VIRGO contains a multitude of non-redundant genes that we
identified in vaginal metagenomes and urogenital bacterial iso-
lates. These non-redundant genes were also clustered into
orthologous groups (VOGs) using a memory-efficient network-
based algorithm that handles nodes connectivity in high dimen-
sionality space[40]. This approach to identifying orthologous pro-
tein sequences allows for great flexibility because it does not rely
on a single sequence similarity cutoff value[41]. These families of
vaginal orthologs will assist the development of a mechanistic
understanding of these proteins and how they relate to health. For
example, van der Veer and co-workers recently identified and
characterized the *L. crispatus* pullulanase (*pulA*) gene which
they show encodes an enzyme with amylase activity that likely
allows this species to degrade host glycogen in the vaginal

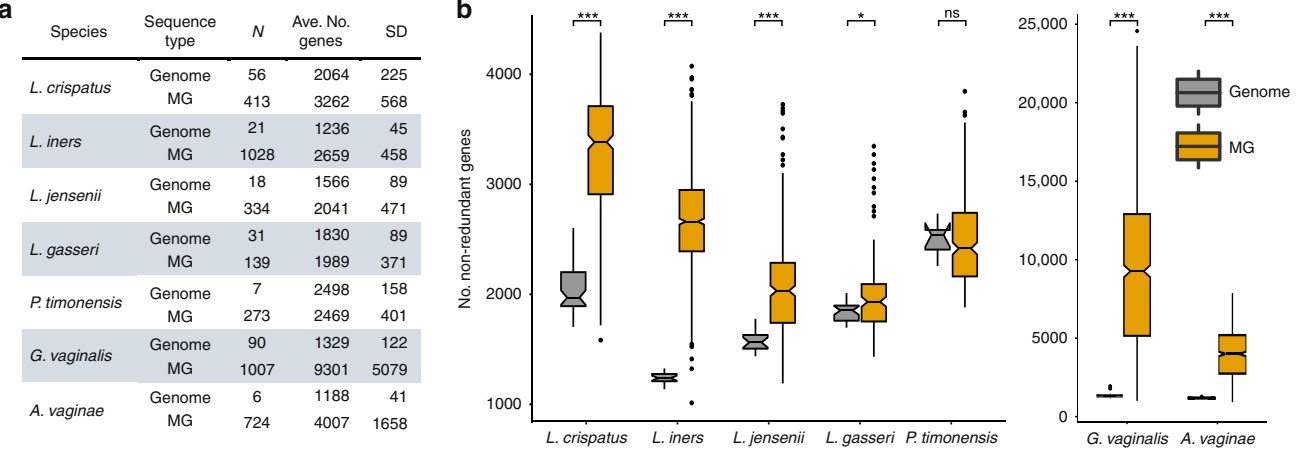

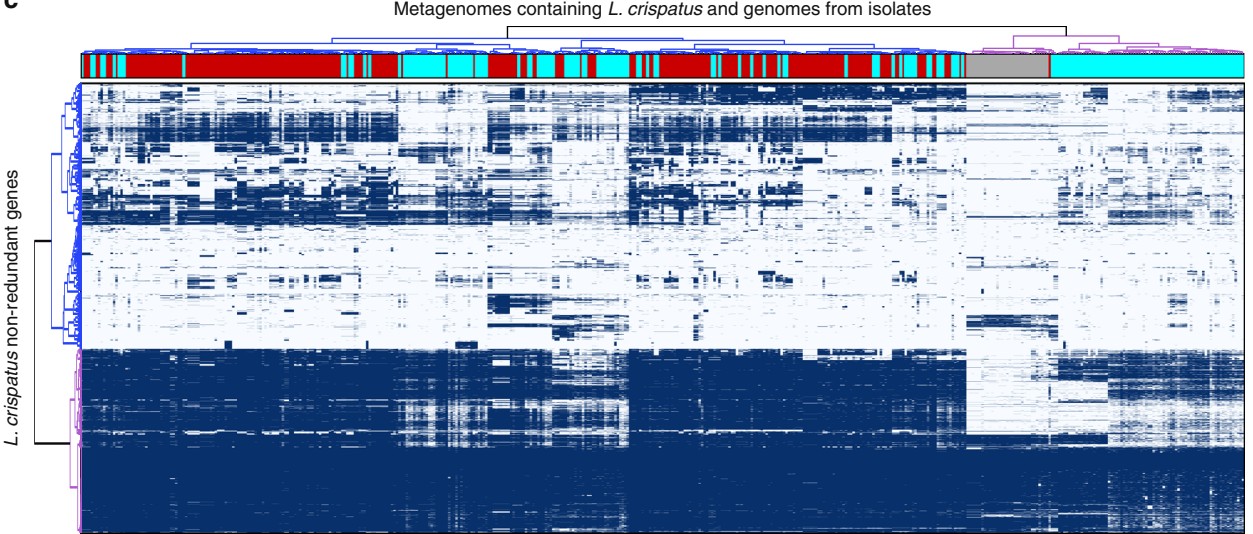

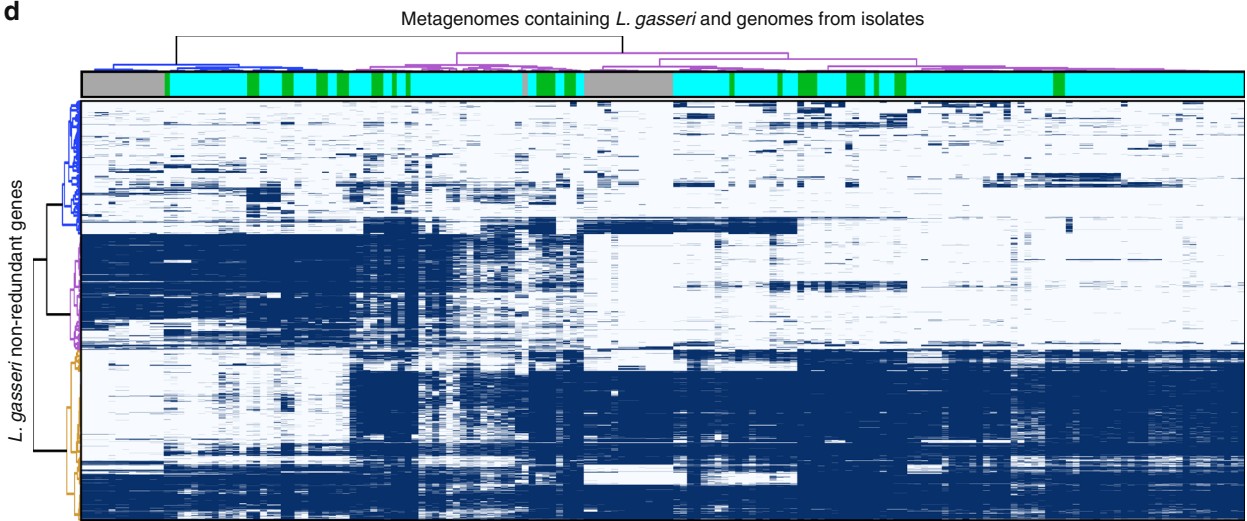

environment[42]. Using VIRGO, we were able to identify pull-ulanase domain containing proteins in 37 other vaginal taxa including: *G. vaginalis*, *L. iners* and *P. timonensis* (Supplementary Data 12), providing insight into the breadth of vaginal bacteria that may be capable of degrading host glycogen. In this way, VIRGO can facilitate knowledge retrieval, hypothesis generation and future experimental validation to advance understanding of vaginal ecosystem.

In our demonstrative analysis of more than 1500 metagenomes, we identified and characterized a wealth of intraspecies diversity that was present within individual vaginal microbial communities. Populations of bacterial species in vaginal communities are likely comprised of multiple strains. Previous studies of the vaginal microbiome have largely treated these species as singular genotypes[43], although some more recent studies have examined intraspecies diversity in these communities[44,45].

**Fig. 6 Intraspecies diversity revealed using VIRGO of seven vaginal species including *L. crispatus*, *L. iners*, *L. jensenii*, *L. gasseri*, and *G. vaginalis*, *A. vaginae* and *P. timonensis*. a** Summary of the number (*N*) of isolate genomes and metagenome (MG) samples with more than 80% of their average genome's number of coding genes for a species, based on a dataset of 1507 in-house vaginal metagenomes characterized using VIRGO. **b** Boxplot of number non-redundant genes in isolate genomes versus vaginal metagenomes. Boxes represent the interquartile ranges, the notch represents the 95% confidence interval, and the line represents the median value. **c** Heatmap of presence/absence of *L. crispatus* non-redundant gene profiles for 56 available isolate genomes (gray) and 413 VIRGO-characterized metagenomes that contained either high (>50% relative abundance, red: *L. crispatus*, green *L. gasseri*) or low (<50% relative abundance, cyan) relative abundance of the species. Hierarchical clustering of the profiles was performed using ward linkage based on their Jaccard similarity coefficient. *number of isolate genomes and metagenome samples. †MG: Metagenomes *$p < 0.05$, ***$p < 0.001$, Student's *t*-test, after correction for multiple comparisons.

Intraspecies diversity is important because it is likely to influence many properties of the communities including their temporal stability and resilience, as well as how they relate to host health. Unfortunately, intraspecies diversity is difficult to detect using typical assembly-based metagenomic analysis strategies, which are notoriously ill suited for resolving strains of the same species[46,47]. VIRGO can be a more suitable tool for characterizing intraspecies diversity because it was built to contain the non-redundant pangenomes of most species common to the vagina. Strict mapping of sequence reads against the VIRGO database provides an accurate and sensitive way of identifying the aggregated non-redundant genes that belong to each species in a metagenome. We expect VIRGO to facilitate future investigations of intraspecies diversity in vaginal microbial communities.

We further showed that, for the seven species we examined, the intraspecies diversity had structure. Vaginal metagenomes from different subjects contained related sets of species-specific non-redundant genes. We postulate that these clusters of samples with shared gene content represent similar collectives of strains which we have termed "metagenomic subspecies". It is expected that, given their shared gene content, these metagenomic subspecies might also share phenotypic characteristics. However, additional studies are needed to characterize differences between metagenomic subspecies and to detail their possible effect on host health. Reconstructing a particular species' metagenomic subspecies might be possible by identifying and combining isolates that adequately cover the genetic repertoire of the metagenomic subspecies. One complication to this approach is that, in many cases, the observed metagenomic subspecies contain non-redundant genes, which have not been observed in isolate genome sequences for that species. This could reflect a limitation in the number of isolate genomes available for a species or even systematic bias in the growth and recovery of species from the vagina[48]. Targeted isolation of strains from communities containing the metagenomic subspecies of interest are needed in order to fill in these gaps in the future.

The value of VIRGO resides in its functions as both a central repository and a highly scalable tool for fast, accurate characterization of vaginal microbiomes. VIRGO is particularly useful for users with limited computational skills, a large volume of sequencing data, and/or limited computing infrastructure. In particular, the metagenome-metatranscriptome data integration enabled by the gene-centric design in VIRGO provides a powerful approach to determine the expression patterns of microbial functions, and in doing so, to characterize contextualized complex mechanisms of host-microbiota interactions in vaginal communities. This feature makes possible the meta-analyses of vaginal microbiome features and the quantitative integration of findings from multiple studies, which helps with the common issue of confounding gene copy number that has been a major challenge in analyzing metatranscriptomic dataset[49]. We also anticipate that VIRGO will be used to process metaproteomic datasets when that practice becomes common and easily accessible. Each of the protein sequence of each gene could be used to map peptides obtained from metaproteomic pipelines and access VIRGO rich annotation. On the other hand, we acknowledge the limitations of the reference based approach of VIRGO. This version is focused on the gene-level de-redundancy and characterization of vaginal microbiome. However, in the future we plan to expand VIRGO to include the capability to identify nucleotide variants within a gene. We believe this will further facilitate our understanding of within-species diversity and evolutionary change in the vaginal ecosystem. The database is primarily focused on bacteria with limited inclusion of viral and fungal gene sequences. Future in-depth profiling of these non-bacterial microbes will allow VIRGO to provide a more complete picture of vaginal microbial communities.

## Conclusion

Efforts are underway to translate our growing understanding of human-associated microbial communities into clinical biomarkers and treatments. A deeper understanding of the complex mechanisms of host–microbiota interactions requires the integration of multi-omics data. VIRGO presents a central reference database and analytical framework to enable the efficient and accurate characterization of the microbial gene content of the human vaginal microbiome. Powered by a rich suite of functional and taxonomic annotations, VIRGO allows for the integrated analysis of metagenomic and metatranscriptomic data. VIRGO further provides a gene-centric approach to describe vaginal microbial community structure including fine scale variation at the intraspecies level. This unprecedented view of intraspecies diversity within a vaginal community is far beyond the scope offered by current genome references. VIRGO is a centralized, and freely available resource for vaginal microbiome studies. It will facilitate the analysis of multi-omics data now common to microbiome studies, and provide comprehensive insight into community membership, function, and ecological perspective of the vaginal microbiome.

## Methods

**Datasets**. Metagenomes used in this study include 211 newly in-house sequenced datasets and 53 vaginal datasets downloaded from the HMP data repository (http://www.hmpdacc-resources.org/cgi-bin/hmp_catalog/main.cgi). Genome sequences of urogenital bacterial isolates deposited in multiple databases were downloaded on November 10, 2016, including GenBank (http://www.ncbi.nlm.nih.gov/), IMG/M: Integrated Microbial Genomes & Microbiomes (https://img.jgi.doe.gov/), and HMP referencing genome database (http://www.hmpdacc-resources.org/cgi-bin/hmp_catalog/main.cgi). After removing duplicate genomes under the same strain names, genomes of 308 urogenital bacterial strains representing 150 bacterial species were included in the catalog. A full list of the genomes and metagenomes used in the construction of the database can be found in Supplementary Data 1. All in-house metagenomic and metatranscriptomic data were derived from vaginal swab or lavage samples collected under Institutional Review Board approved protocols, and after obtaining written informed consent from all the participants.

**Nucleic acid extraction, library construction, and metagenome and metatranscriptome sequencing**. The included 211 in-house metagenomes were generated as follows: whole genomic DNA was extracted from 300 μl aliquot of vaginal ESwab re-suspended into 1 ml Amies transport medium (ESwab, Copan Diagnostics Inc.) and preserved at −80 °C. DNA was isolated as previously described[21].

Briefly, cells were lysed with the addition of an enzymatic cocktail: 50 μl of lyzosyme (10 mg/ml), 6 μl of mutanolysin (25,000 U/ml Sigma- Aldrich), 3 μl of lysostaphin (4000 U/ml in sodium acetate; Sigma- Aldrich), and 41 μl of TE50 buffer (10 mM Tris·HCL and 50 mM EDTA, pH 8.0). Mixture was incubated for 1 h at 37 °C and then disrupted by bead beating (0.1-mm-diameter zirconia/silica beads, BioSpec Products) for 1 min at room temperature in a Mini-Beadbeater-96 (2100 rpm, BioSpec Products). DNA was isolated from the resulting crude lysate using the QIAamp DNA Mini Kit (Qiagen) and samples were eluted with $2 \times 200$ μl of TE buffer (10 mM Tris-Cl, 0.5 mM EDTA; pH 9.0). DNA concentrations was measured using the Quant-iT PicoGreen dsDNA assay kit (Invitrogen). The shotgun metagenomic sequence libraries were constructed from the extracted DNA using Illumina Nextera XT kits and sequenced on an Illumina HiSeq 2500 platform (150 bp paired end mode, eight samples per lane) at the Genomic Resource Center at the University of Maryland School of Medicine.

The metatranscriptomes used to demonstrate the use of VIRGO for the analysis of community-wide gene expression were obtained from RNA extracted from vaginal swabs stored in 2 ml Amies Transport Medium-RNAlater solution (50/50%, vol/vol) archived at −80 °C. A total of 500 μl of ice-cold PBS was added to 1000 μl of that solution and spun down at 8,000×g for 10 min. The pellet was resuspended in 500 μl ice-cold RNase-free PBS with 10 μl β-mercaptoethanol. The suspension was transferred to Lysis Matrix B tube (MP Biomedicals) containing 100 μl 10% SDS and 500 μl acid phenol and beads beaded using a FastPrep instrument (MP Biomedicals) for 45 s at 5.5 m/s. The aqueous phase was mixed with 250 μl acid phenol and 250 μl 24:1 chloroform:isoamyl alcohol. The aqueous layer was again transferred to a fresh tube and mixed with 500 μl 24:1 chloroform: isoamyl alcohol. For every 300 μl resulting aqueous solution, we added 30 μl of 3 M sodium acetate, 3 μl of glycogen (5 mg/ml), and three volumes of 100% ethanol. The mixture was incubated at −20 °C overnight to precipitate the nucleic acids. After centrifugation at 13,400×g for 30 min at 4 °C, the resulting pellet was washed, dried, and dissolved in 100 μl of DEPC-treated water. Carryover DNA was removed by: (1) treating twice with Turbo DNase free (Ambion, Cat. No. AM1907) at two half-hour intervals according to the manufacturer's protocol for rigorous DNAse treatment, (2) purifying twice using gDNA-eliminator columns (QIAGEN) before and after DNase treatment followed by RNeasy column purification (QIAGEN). We further conducted PCR using 16S rRNA primer 27 F (5′-AGAGTTTGATCCTGGCTCAG-3′) and 534 R (5′-CATTACCGCGGCTGCTGG-3′) to confirm DNA removal. The quality of extracted RNA was checked using an Agilent 2100 Expert Bioanalyzer Nano chip. Ribosomal RNA removal was performed according to the manufacturer's protocol of a combined Gram-positive, Gram-negative and Human/mouse/rat Ribo-Zero rRNA Removal Kit (Epicenter Technologies). The resulting RNA was purified using Zymo RNA clean & Contentrator-5 column kit (ZYMO Research). RNA final quality was checked using an Agilent RNA 6000 Expert Bioanalyzer Pico chip. Sequencing libraries were prepared using the TruSeq RNA sample prep kit (Illumina) following a modification of the manufacturer's protocol: cDNA was purified between enzymatic reactions and library size selection was performed with AMPure XT beads (Beckman Coulter Genomics). Library sequencing was performed using the Illumina HiSeq 2500 platform (150 bp paired end mode, eight samples per lane).

### Construction of the human vaginal non-redundant gene catalog (VIRGO).

Multiple bioinformatics pre-processing steps were applied to the raw shotgun metagenomic sequence datasets, including (1) eliminating all human sequence reads (including human rRNA LSU/SSU sequence reads) using BMTagger v3.101[50] against a standard human genome reference (GRCh37.p5[51]), (2) in silico microbial rRNA sequence reads depletion by aligning all reads using Bowtie (v1)[52] against the SILVA PARC ribosomal-subunit sequence database (v132)[13] to eliminate misassemblies due to these repeated regions. After each of these steps, the paired reads were removed; (3) stringent quality control using Trimmomatic v0.36[53], in which the Illumina adapters were excised, reads were trimmed using a 4 bp sliding window with an average quality score threshold of Q15, and reads containing any ambiguous bases were removed. MetaPhlAn (v2)[19] was subsequently used to establish taxonomic profiles after these pre-processing steps. Samples were then clustered in community state types (CSTs) using taxa abundance tables and the Jensen-Shannon divergence metrics as previously described[21]. Species accumulation curves and diversity estimates for rarefied samples were computed using R package iNEXT v2.0[54] and vegan v2.5-5[55]. The 264 vaginal metagenomes were then assembled using IDBA-UD (v1.0)[56] with a k value range of 20–100. Genes were called on the resulting contigs using MetageneMark (v3.25)[25] to predict CDSs with the default settings. Genes and gene fragments that were at least 99 bp long, with greater than 95% identity over 90% of the shorter gene length were clustered together by a greedy pairwise comparison implemented in CD-HIT-EST (v4.6)[57], according to the clustering procedure and threshold defined previously[12,15]. The gene with the longest length ≥99 bp was used as the representative for each cluster of redundant genes.

### Taxonomic and functional annotations of VIRGO.

The non-redundant genes were annotated with a rich set of taxonomic and functional information. Genes that originated from an isolate sequence genome were automatically assigned that species name. For metagenomes, taxonomy was assigned to a metagenomic contig by mapping the sequence reads making up that contig to the Integrated Microbial Genomes (IMG) reference database (v400) using bowtie (v1, parameters: "-l 25–fullref–chunkmbs 512–best–strata -m 20"). A secondary filter was applied so that the total number of mismatches between the read and the reference was less than 35, and that the first 25 bp of the read matched the reference. Using the results of this mapping, taxonomy was assigned to all genes encoded on the contig that met the following four criteria: (1) at least 95% of the reads mapped to the same bacterial species, (2) the remaining 5% off-target reads did not map to a single species, (3) the contig had at least 2× average coverage and >50 reads, (4) at least 25% of the contig length had reads mapped onto. These stringent criteria were used to ensure high fidelity of the taxonomic assignments and a low contribution of potentially chimeric contigs. To further diminish the risk of incorporating false taxonomic assignments, the annotations of the contigs belonging to species at low relative abundance in the sample were removed. Genome completeness was estimated as the fractional representation of the genome in the metagenome using BLASTN (minimal overlapping >60% of the shorter sequence and >80% sequence similarity). For each metagenome, only taxonomic assignments originating from species with at least 80% representation were incorporated. The genes that shows >80% sequence similarity over 60% of query gene length to the non-redundant genes were then assigned. The non-redundant genes in VIRGO were searched against fungal database that includes 5 vaginal yeast species in 40 genomes (listed in Supplementary Data 5) using BLASTN, that a gene must have at least 80% sequencing similarity with over 60% overlapping length to be curated. We also annotated potential phage genes that may be present in VIRGO by searching against phage orthologous groups or Prokaryotic virus orthologous groups (version 2016)[41], using BLASTN and included the ones at >80% sequence similarity over 60% of query gene length in annotation (Supplementary Data 5). Functional annotations based on the standard procedure for each of 17 functional databases, including: cluster of orthologous groups (COG (v1)[58], eggNOG (v4.5)[59], KEGG (FTP Release 2013-03-18)[60], conserved protein domain (CDD (v3.14)[61], Pfam (v30.0)[62], ProDom (v20.119)[63], PROSITE (V20.119)[64], TIGRFAM (v15.0)[65], InterPro (v60.0)[66], domain architectures (CATH-Gene3D (v4.1)[67], SMART (v7.1)[68], intrinsic protein disorder (MobiDB (v2.0)[69], high-quality manual annotation (HAMAP (v201605.11)[70], protein superfamily (PIRSF (v3.01)[71], a compendium of protein fingerprints (PRINTS[72]), and gene product attributes (Gene Ontology, JCVI SOP[35]).

### Construction of vaginal orthologous groups (VOGs) for protein families.

The non-redundant genes were also clustered based on orthology to generate a set of VOGs. To do this we used a modified version of a Jaccard clustering method previously implemented[36]. We performed an all-versus-all BLASTP search among the translated coding sequences (CDS) of the non-redundant genes included in VIRGO[73]. The all-against-all BLASTP matches was used to compute Jaccard similarity coefficient for each pair of translated CDSs, without constraints based on which sample or microorganism from which it originated. Only BLASTP matches with 80% sequence identity and 70% overlap, and an E-value less than 1E−10 were used in the calculation of the Jaccard similarity coefficient. The filtered BLASTP results were then used to define connections between pairs of translated CDSs resulting in a network graph with the translated CDSs as nodes and their connections as edges. The Jaccard similarity coefficient was then calculated as the number of nodes that had direct connections to the two translated CDSs divided by the total number of nodes that had direct connections to either of the two translated CDSs in the network (intersection and union)[74]. JACs were defined as a set of translated CDSs whose Jaccard similarity coefficient was at least 0.55. If two translated CDSs from different JACs were reciprocal best matches according to the BLASTP searches, the two JACs were merged. Finally, the alignment program T-Coffee[75] was used to assess the alignment quality within the JACs and to calculate the alignment score. The sequences, alignment, and phylogenetic trees for each of the JOCs are available at www.virgo.igs.umaryland.edu.

### Bioinformatics analysis.

The comprehensiveness of VIRGO was tested using vaginal metagenomic data from vaginal metagenomes of North American women not including in the construction of VIRGO and sequenced in this study, as well as women from African[22], and China[23]. The sequences reads were first pre-processed as described above for VIRGO construction and mapped to the VIRGO contigs using bowtie (v2; parameters:–threads 4–sensitive-local -D 10 -R 2 -N 0 -L 22 -i S,1,1.75 -k 1–ignore-quals–no-unal)[76], according to the criteria used previously in the construction of a gut gene catalog[12]. Any unmapped reads were compared to the GenBank nt database[77] using BLASTN and an E-value of 1E−10 as cutoff. To annotate BVAB1 genes in VIRGO, we used BLASTN and an E-value of 1E−10 as cutoff, the matched genes with percent identity >95% over >90% of gene length were annotated as BVAB1 genes. To retrieve pullulanase (pulA) genes in VIRGO, we used conserved protein domain CDD[61] annotation and keyword "pullulanase". To further demonstrate the comprehensiveness of VIRGO and that VIRGO captures the pangenome of selected species, species specific metagenome accumulation curves for the number of non-redundant genes were constructed for seven vaginal species by rarefaction with 100 bootstraps: L. crispatus, L. iners, L. jensenii, L. gasseri, and G. vaginalis, A. vaginae and P. timonensis.

### Using VIRGO to characterize within community intraspecies diversity.

Intraspecies diversity analyses were conducted by mapping isolate genome

sequences as well as vaginal metagenomes to VIRGO. We chose to focus our analysis on the previously mentioned seven vaginal species. Accession numbers for genomes of the four *Lactobacillus* species (*L. crispatus, L. iners, L. jensenii,* and *L. gasseri)* and three additional species (*G. vaginalis, A. vaginae* and *P. timonensis*) can be found in Supplementary Data 11. A total of 1507 vaginal metagenomes including 1403 in-house from de-identified vaginal swab and lavage specimens and 76 publicly available, were mapped against VIRGO. These metagenomic datasets are available at www.virgo.igs.umaryland.edu. For each of the seven species, a presence/absence matrix for the species' non-redundant genes (at least 3 reads mapped onto the gene) was constructed that included the data from species' isolate genomes and all metagenomes that contained at least 80% of the average number of genes encoded on a genome of that species. Comparisons of the number of non-redundant genes present in the species isolate genomes versus the metagenomes in which they appeared where conducted using student *t*-test. Hierarchical clustering was performed on the boolean matrix of the species' non-redundant genes using Jaccard clustering implemented in the *vegan* package in R. A tutorial describing how to use VIRGO and VOG is available online at https://github.com/ravel-lab/VIRGO.

**Reporting summary**. Further information on research design is available in the Nature Research Reporting Summary linked to this article.

## Data availability

The VIRGO non-redundant nucleotide gene database, VOG amino acid protein family database, curated taxonomy and functional information, detailed tutorials and the metagenomes used in the analyses are available at http://virgo.igs.umaryland.edu.

## Code availability

All code used in this study are publicly available at https://github.com/ravel-lab/VIRGO. This includes the Jaccard index clustering code.

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

## Acknowledgements

The authors thank Drs. Douglas Kwon and Matthew Hayward for their helpful assistance in analyzing African women metagenomes. The authors thank Dr. Nan Qin and Qian Xu for their helpful assistance in analyzing Chinese women metagenomes. Research reported in this publication was supported by the National Institutes of Allergy and Infectious Diseases and Nursing Research of the National Institutes of Health under award numbers U19AI084044, R01NR015495 and R01AI116799, and the Bill & Melinda Gates Foundation award OPP1189217.

## Author contributions

B.M., J.R. designed the research. B.M., M.T.F., J.H., and J.R. performed the research. B.M., M.S.H. generated the data. B.M., M.T.F., J.H., and J.C. analyzed the data. B.M., M.T.F., R.M.B., and J.R. interpreted the data and wrote the paper.

## Competing interests

J.R. is co-founder of LUCA Biologics, a biotechnology company focusing on translating microbiome research into live biotherapeutics drugs for women's health. All other authors declare that they have no competing interests.
