## [Peer Review File · Nature Communications]

Reviewers' comments:

Reviewer #1 (Remarks to the Author):

The manuscript is a nice compilation of public and newly sequenced genomic and transcriptomic data for the vaginal microbiome. The effort to make a curated, well-annotated resource will certainly be appreciated by the research community. The authors describe several interesting and novel information about the microbial constitutes based on the meta-genomic information that puts this the paper a step above just a database resource.

The data compilation approach used is reasonable and justifiable. Of course, some assumptions were made for the data curation that will affect the downstream interpretation, but this is the case with any compiled dataset.

The largest issue I have was the installation and guidance for using the resource. The GitHub repo provided ([GitHub - Ravel-Laboratory/VIRGO](https://github.com/Ravel-Laboratory/VIRGO)) is very bare-bones in terms of workflow description, instructions, tutorial support, and description of the different components of the database. This could affect appeal and usability for researchers in the field, especially when modern expectations of Markdown/HTML tutorials and informative content.

Reviewer #2 (Remarks to the Author):

The scale and breadth of the data and the analyses carried out by the authors are very impressive. Databases like the one described in this manuscript are very informative and will very likely to be utilized by numerous studies. However, there is insufficient details in certain sections that need to be improved. There are several vaginal relevant taxa that were not included in the database. The overall writing and flow of the paper are not consistent, a careful review is recommended.

Line 43 – replace period with comma and lowercase ‘Yet’

Line 45-48 – revise sentence structure

Line 123 – Is there a reason for this difference?

Line 130 – Is this finding due to larger number of reads? Better read-quality? Newer technologies?

Line 135/Suppl. Table 1 – worksheet #4 (Suppl. Table 1) entitled ‘List of genomes isolated from vagina, reproductive or urinary system deposited in GenBank’ contains genomes isolated from ‘environmental swabs’ as indicated for instance in ncbi taxon ids 910413 and 910414. Please clarify or make appropriate corrections. GenBank Assembly ID of the genomes used to build the database should be provided.

The authors provided a vast list of genomes, however there are many vaginal relevant genomes that were not included such as *Sneathia amnii*, *Sneathia sanguinegens*, *Candidatus Mycoplasma girerdii*, *Mycoplasma hominis* and BVAB1. These organisms are usually associated with BV, preterm birth, etc. It seems like a missing opportunity not to include them. Lack of such important taxa is problematic.

Suppl. Table 2. Samples SRS014494, SRS015072, SRS044742, SRS022158 and SRS049237 are predominantly *L. crispatus* (47% or higher) however they are classified as CST-IV-B. In addition, the majority (80%) of the samples classified as CST IV-A however, are predominantly composed by *G. vaginalis* (at least 50% relative abundance). Historically, CST IV-A is described by “modest proportions” of either *L. iners*, *L. crispatus* or other *L. spp* + various species of anaerobic bacteria) and CST IV-B is usually associated with higher proportions of *Atopobium*, *Prevotella*, *Sneathia*, etc. With the addition of different populations and different cohorts (bv, preterm, hiv, etc) non-Lactobacillus taxa are also being found to be a dominant type and to compose probably several different types. There is a need to update this to reflect the diversity across populations/cohorts

Line 137 – ‘Taxonomic analysis’. Which tool was used to perform taxonomic classification?

Line 141 – please provide full scientific name, since this is the first time in the text. *G. vaginalis*, *A. vaginae*, *P. amnii*, *P. timo*

Line 157 – ‘91 vaginal metagenomes..’ Suppl Table 1 details only the 211 metagenomes that were used to build the database, what are these 91 metagenomes subset used to validate VIRGO? A brief description (including readset stats) of the African and Chinese women cohorts would be helpful.

Line 161 – which ‘subsets of the sequence contigs’ are the authors referring to?

Line 169 – This seems quite unusual, 12% of unmapped reads! Why not blastx?

Line 172 – This statement is a quite simplified view of the vaginal microbiome. Grosmann’ study shows contrasting differences. 58% of the women have a community type (probably 2 or 3 different community types) characterized by a non-Lactobacillus dominant communities.

Cohorts of women infected with HIV, at high risk for preterm birth, African descendants are less likely to have Lactobacillus-dominated communities.

Line 201 – Supp Table 4 shows 268 unique species

Line 204 – There is a BVAB1 genome available from a recent publication, Fettweis et al., 2019

Line 205 – eliminate ‘been’

Line 205/206 – This statement has to be updated. ‘BVAB1 was only been previously detectable using a

partial 16S rRNA gene reference sequence’ There are two publications which BVAB1 genome was used

to identify metagenome and metatranscriptome reads. (Fettweis et al., 2019 and Serrano et al., 2019)

Line 207 – Revise sentence: ‘It was found abundantly present in most of the metagenomes with a prevalence...’

Line 224 – It is not clear (Methods) how the accumulation curves were generated. Please clarify how saturation was determined? I would argue that *A. vaginae* is approaching saturation but it doesn’t seem to have reach it yet. Would extrapolation curves using subsampling (e.g. 1%, 10%) be more appropriate?

Line 231 – insert comma after ‘saturation’

Line 233 – please revise: ‘The non-redundant genes were decorated with a rich set of functional annotations’. ‘decorated’ seems an unusual choice.

Line 234 – eliminate ‘both’

Line 247 – replace ‘interrogate’ with ‘investigate’

Line 246-247 – this sentence can be eliminated, similar sequence in Line264-266

Line 248 – eliminate ‘Briefly’

Line 261 – Files should be available

Line 271 – replace ‘a’ with ‘an’

Line 271 – break sentence between ‘vaginolysin’ and ‘one’

Line 272 – eliminate ‘had’

Line 276 - revise beginning of sentence: ‘As another example to use VOG for a large-scale data mining of protein family of interest,...’

Line 308-317 – ‘L. iners-dominant communities...’ Are the authors referring to CST III only? If not, please describe how dominant was defined? This finding is not very surprising. Unlikely L. crispatus, L. iners seems to be found, more often than not, in microbiomes profiles with very diverse community composition. This would explain the classification into the HGC group and also the difference observed between LGC and HGC. Similar to L. iners, G. vaginalis can found as a dominant community (more than 10% of the metagenomic samples have 80% or more G. vaginalis) as well as part of a very diverse community.

Line 329-33 – Is there a statistical analysis to support this?

Line 340 – Use *Mageibacillus indolicus* instead of BVAB3

Line 341-345 – Please revise sentence for clarity

Line 363-364 – ‘(TB3)’ should immediately follow ‘BV episode’

Line 378 – There is no details regarding the 1507 metagenome samples. Please provide the accession numbers. Are these samples part of a separate manuscript? Please provide citation.

Line 382 – make it possessive ‘species’

Line 394 – eliminate ‘basically’

Line 411 – revise sentence

Line 414 – what well-established practices?

Line 438-441 – revise sentence

Line 445 – replace ‘interrogation’ with ‘investigation’

Line 554 – replace ‘referenced’ with ‘reference’

Line 587-588 – This sentence seems dubious ‘416 urogenital bacterial strains and 321 bacterial species’. There are ~146 bacterial species, please clarify.

Line 596 – Eliminate “Briefly”

Line 631 – Please explain “Sequencing libraries of A and B containing 6 bp..”

Methods – Please describe read length, etc... how many samples per lane, Q15 seems to low

Line 725-727 – BVAB1 is not listed in Suppl Table 1, please review this sentence for details and clarification. which database was used to annotate BVAB1 genes? Why there is a different pipeline for BVAB1?

Line 755 – ‘Their accession numbers can be found in Supplementary Table 12.’ Suppl. Table 12 has Taxonomic distribution of pullulanase domain-containing proteins included in VIRGO.

Line 1175 - Figure 3. How the genome size of BVAB1 was determined?

Reference 11 is not cited. Remove randomly capitalized words.

Legend Figure 6d. At the top bar, what are the colors blue, green and gray representing?

Title of Suppl Table 3 indicates ‘212 in-house’

Supp Table 4 (multiple tables and throughout the manuscript) – Use *Mageeibacillus indolicus* instead of BVAB3

Lachnospiraceae_bacterium (row 225), there are many members of this family, please specify.

Don’t italicize family name, Lachnospiraceae. Column C refers to genera, Lachnospiraceae is not a genus.

Reviewer 1's comment:

“The manuscript is a nice compilation of public and newly sequenced genomic and transcriptomic data for the vaginal microbiome. The effort to make a curated, well-annotated resource will certainly be appreciated by the research community. The authors describe several interesting and novel information about the microbial constitutes based on the meta-genomic information that puts this the paper a step above just a database resource.

The data compilation approach used is reasonable and justifiable. Of course, some assumptions were made for the data curation that will affect the downstream interpretation, but this is the case with any compiled dataset.

The largest issue I have was the installation and guidance for using the resource. The GitHub repo provided ([GitHub - Ravel-Laboratory/VIRGO](https://github.com/Ravel-Laboratory/VIRGO)) is very bare-bones in terms of workflow description, instructions, tutorial support, and description of the different components of the database. This could affect appeal and usability for researchers in the field, especially when modern expectations of Markdown/HTML tutorials and informative content.”

Reviewer 2's comment:

“The scale and breadth of the data and the analyses carried out by the authors are very impressive. Databases like the one described in this manuscript are very informative and will very likely to be utilized by numerous studies. However, there is insufficient details in certain sections that need to be improved. There are several vaginal relevant taxa that were not included in the database. The overall writing and flow of the paper are not consistent, a careful review is recommended.”

Note: In order for the manuscript to comply with the journal guidelines, it was modified as follows: 1/ the abstract was shortened to 150 words; 2/ the main text was shortened to less than 5,000 words (Introduction, Results, Discussion), as a result, a new supplementary information file 1 was added to the manuscript that describes the results and discussion on gene richness definition of the vaginal microbiome; 3/ the headers were shortened to be 60 characters or less and 4/ the number of references were reduced to 70.

Response to general comments:

The reviewers raised three main points which we have thoroughly addressed in the revised manuscript: 1/ the availability of VIRGO and its associated files (reviewer #2); 2/ The usability of VIRGO (reviewer #1); and 3/ a request to add a few recently released vaginal bacteria genomes. We have addressed these points as follows:

1/ We have set up a download website for VIRGO on our institutional webserver (<http://virgo.igs.umaryland.edu/>), where all associated files can be downloaded (including all the

metagenomes that were used to build the gene catalogue as requested by reviewer #2). This public, permanent link is hosted by the Institute for Genome Sciences at the University of Maryland School of Medicine and contains: 1) the 1,507 de-identified vaginal metagenomes used to build VIRGO; 2) the complete genome sequence of BVAB1 in GenBank format; 3) The sequences, alignment, and phylogenetic trees for each of the VIRGO orthologous protein families (VOG); 4) the updated VIRGO database and annotation. All of these can be freely downloaded without any restrictions. We found this option to be the most efficient because of the large size of the datasets, as fast download of these large data files is provided through an Aspera server.

2/ To address usability, we have included a detailed tutorial at <https://github.com/ravel-lab/VIRGO>. The availability of this tutorial should facilitate the adoption of VIRGO among researchers with different computational expertise. This updated Markdown tutorial, along with the workflow description, instructions, tutorial support, and description of different components of the database are now available. The tutorial was extensively tested by researchers with limited expertise in computational bioinformatics and received enthusiastic feedback for its comprehensiveness and ease of use. A link to virgo.igs.umaryland.edu is now provided on the GitHub site.

3/ We have expanded VIRGO taxonomic annotations by adding the few recently released and important vaginal bacteria, including: BVAB1 (*Candidatus Lachnocurva vaginae* (BVAB1), *Sneathia amnii*, *Sneathia sanguinegens*, and *Mycoplasma hominis*. We strongly agree with reviewer #2 that it is important that these taxa be included in VIRGO. Given that VIRGO was built using metagenomic data, the genetic contents of the vast majority of vaginal bacterial taxa are already present in the database even if they have not been taxonomically annotated. To ensure the quality of VIRGO's taxonomic annotations, we only included those which originated from isolate genome sequences but not metagenome-assembled genomes that are not closed (e.g. *Candidatus Mycoplasma girerdii* was not included).

Response to minor comments of Reviewer 2:

Line 43 – replace period with comma and lowercase ‘Yet’

We have made the suggested change.

Line 45-48 – revise sentence structure

We have change the sentence which now reads: “Taxonomic profiling of the human microbiome via 16S rRNA gene amplicon sequencing has provided critical insight into the potential role of the microbiota in a wide array of common diseases, yet these data routinely fall short of describing their etiology, including bacterial vaginosis², Crohn’s disease³ or psoriasis⁴, among others.”

Line 123 – Is there a reason for this difference?

It is not clear to us what causes the differences in the number of human reads associated with *Lactobacillus*-dominated metagenomes compared to those deficient in *Lactobacillus*. We report this difference but feel identifying the potential cause to be outside the scope of this study.

Line 130 – Is this finding due to larger number of reads? Better read-quality? Newer technologies?

This point is now discussed. We believe this is due to the lack of genetic diversity in the HMP vaginal metagenomes, which are from women with microbiota almost exclusively dominated by *Lactobacillus*.

Line 135/Suppl. Table 1 – worksheet #4 (Suppl. Table 1) entitled ‘List of genomes isolated from vagina, reproductive or urinary system deposited in GenBank’ contains genomes isolated from ‘environmental swabs’ as indicated for instance in ncbi taxon ids 910413 and 910414. Please clarify or make appropriate corrections. GenBank Assembly ID of the genomes used to build the database should be provided.

The list of genomes was retrieved from genBank on Nov 9th, 2015. They were retrieved based on the following criteria of the metadata provided on GenBank report: 1) The ecosystem is “host-associated” and the ecosystem subtype is “vagina”; 2) If no ecosystem subtype was specified, ecosystem type is listed as “reproductive system”; 3) If neither ecosystem or ecosystem subtypes was specified, the habitat contain the key word “vagina”. We used these criteria to retrieve genomes because the genBank metadata is not always in a consistent format and we tried to include a comprehensive, unbiased set of genomes associated with urogenital system. The *Salmonella* genomes were not removed from our list because they met the selection criteria. VIRGO results are not affected by the inclusion of these *Salmonella* genomes because, unless the species is present reads will not be mapped onto them. In our analysis of 1,507 metagenomes no *Salmonella* genes were identified.

The authors provided a vast list of genomes, however there are many vaginal relevant genomes that were not included such as Sneathia amnii, Sneathia sanguinegens, Candidatus Mycoplasma girerdii, Mycoplasma hominis and BVAB1. These organisms are usually associated with BV, preterm birth, etc. It seems like a missing opportunity not to include them. Lack of such important taxa is problematic.

We have now expanded VIRGO taxonomic annotations by adding the few recently released vaginal bacteria, including: BVAB1 (*Candidatus Lachnocurva vaginae* (BVAB1)), *Sneathia amnii*, *Sneathia sanguinegens*, and *Mycoplasma hominis*. Given that VIRGO was built from metagenomic data, the genetic contents of the vast majority of vaginal bacterial taxa were already present in the database even if they had not been taxonomically annotated. To ensure the quality of VIRGO’s taxonomic annotations, we only included genome sequences from organisms that originated from isolate but not metagenome-assembled genomes, thus *Candidatus Mycoplasma girerdii* was not included.

Suppl. Table 2. Samples SRS014494, SRS015072, SRS044742, SRS022158 and SRS049237 are predominantly L. crispatus (47% or higher) however they are classified as CST-IV-B. In addition, the majority (80%) of the samples classified as CST IV-A however, are predominantly composed by G. vaginalis (at least 50% relative abundance). Historically, CST IV-A is described by “modest proportions” of either L. iners, L. crispatus or other L. spp + various species of anaerobic bacteria) and CST IV-B is usually associated with higher proportions of Atopobium, Prevotella, Sneathia, etc. With the addition of different populations and different cohorts (bv, preterm, hiv, etc) non-Lactobacillus taxa are also being found to be a dominant type and to compose probably several different types. There is a need to update this to reflect the diversity across populations/cohorts

Our motivation for using CST assignments was to demonstrate that we had included many samples which were not dominated by *Lactobacillus* when we built VIRGO. As pointed out by the reviewer, VIRGO would not be very useful if it did not represent the diversity of communities that are found in the human vagina. The CST assignments were performed using Ward linkage hierarchical clustering based on Jensen-Shannon divergence metrics, described previously

(https://www.pnas.org/content/108/Supplement_1/4680). Hierarchical clustering can yield non-optimal assignments when the dataset is small (N=264). We have made the changes suggested by the reviewer to recategorize five samples with a moderate relative abundance of *L. crispatus* to from CDT IV to CST I. In Supplemental table 2 we have adjusted the naming scheme for IV-A and IV-B as suggested by the reviewer so as to conform with previous studies.

Line 137 – ‘Taxonomic analysis’. Which tool was used to perform taxonomic classification?

We used Metaphlan V2 to accomplish this task. We have added this information to the manuscript. Which now reads:

“Taxonomic analysis of the metagenomes included in VIRGO using MetaPhlAn (v2)¹⁵, revealed that these communities contained 312 bacterial species present in $\geq 0.01\%$ relative abundance (Supplementary Table 2).”

Line 141 – please provide full scientific name, since this is the first time in the text. G. vaginalis, A. vaginae, P. amnii, P. timo

We have added the genus and species designations for these bacteria upon their first mention throughout the text.

Line 157 – ‘91 vaginal metagenomes..’ Suppl Table 1 details only the 211 metagenomes that were used to build the database, what are these 91 metagenomes subset used to validate VIRGO? A brief description (including readset stats) of the African and Chinese women cohorts would be helpful.

The African and Chinese cohorts and their respective metagenomic datasets were described in their original studies which we have cited (now ref 18 and 19).

Line 161 – which ‘subsets of the sequence contigs’ are the authors referring to?

We have revised this sentence, which now reads:

“Reads from these metagenomes were mapped to the complete set of sequence contigs used to build VIRGO as well as different subsets of these contigs based on the dataset from which they originated (Fig. 1).”

The subsets are described in Figure 1 which is now referred to at the end of the sentence. Briefly, we separated the contigs used to build VIRGO by their origin (metagenome versus genome, *in-house* versus HMP DACC).

Line 169 – This seems quite unusual, 12% of unmapped reads! Why not blastx?

We believe that blastx is not the best tool for this analysis because we are mapping the short reads against contigs which contain both coding and noncoding regions. As described in the manuscript, Bowtie was used to accomplish this task which is a standard tool for short-read mapping that affords both accuracy and recall. It is not unexpected that a larger portion of reads might fail to map from the Chinese cohort. VIRGO still captures the vast majority of these data (88%). This could result from 1) differences in the species/strains that make up some of their bacterial communities 2) differences between the Chinese human genome and the reference genome we used to remove human reads 3) contamination in the Chinese metagenomes from some unknown source. As we did not sequence these metagenomes we cannot guarantee their integrity and quality. 12% of unmapped reads is still larger than what is obtained with current genome sequence databases such as GenBank. The possibility of poor quality and potential contamination is highlighted by the fact that among the reads that did not match to VIRGO from Chinese women (these 12%), 99.9% failed to match anything in GenBank (See Supplementary Figure 4).

Line 172 – This statement is a quite simplified view of the vaginal microbiome. Grosmann’ study shows contrasting differences. 58% of the women have a community type (probably 2 or 3 different community types) characterized by a non-Lactobacillus dominant communities. Cohorts of women infected with HIV, at high risk for preterm birth, African descendants are less likely to have Lactobacillus-dominated communities.

The statement “It further shows that the bacterial genetic diversity in the vaginal microbiome across populations is somewhat homogenous.” refers to the amount of genomic diversity found in vaginal microbiome at the population level. The fact that African and Chinese women metagenomes map to such high percentage to VIRGO indicates that most of the genomic diversity of the vaginal microbiome is captured in VIRGO (which is built from North American women), thus across population there is overall very little genomic diversity, hence the statement about homogeneous genetic diversity. The statement does not refer to difference observed in microbiota types (non-*Lactobacillus* dominated versus *Lactobacillus* dominated) across population which we have reported previously.

Line 201 – Supp Table 4 shows 268 unique species

Supplementary Table 4 was corrected and now list the correct 271 taxa.

Line 204 – There is a BVAB1 genome available from a recent publication, Fettweis et al., 2019

Unfortunately, this genome was not available when we submitted the manuscript, however, we have now edited the main text to include mention of the availability of this genome (references 23 and 24).

Although, we have annotated BVAB1's gene contents in VIRGO using a fully closed BVAB1 genome, and six additional MAGs that we have sequenced and are in the process of publishing (under review).

This genome is available at www.virgo.igs.umaryland.edu.

Line 205 – eliminate 'been'

We have corrected this typo.

Line 205/206 – This statement has to be updated. 'BVAB1 was only been previously detectable using a partial 16S rRNA gene reference sequence' There are two publications which BVAB1 genome was used to identify metagenome and metatranscriptome reads. (Fettweis et al., 2019 and Serrano et al., 2019)

These two studies had not been published when we submitted our manuscript. As suggested, we have revised this sentence to reference these two studies. The sentence now reads:

“This includes BVAB1, an as of yet unculturable vaginal species, for which a closed genome and several metagenome-assembled genomes (MAGs) were recently made available²²⁻²⁴.”

Line 207 – Revise sentence: 'It was found abundantly present in most of the metagenomes with a prevalence...'

We have revised this sentence as suggested by the reviewer. “We found this species to be a common member of the vaginal microbiota with a prevalence of 15.6% and a mean relative abundance of 18.9% +/- 0.01 as shown in **Figure 3D**.”

Line 224 – It is not clear (Methods) how the accumulation curves were generated. Please clarify how saturation was determined? I would argue that A. vaginae is approaching saturation but it doesn't seem to have reach it yet. Would extrapolation curves using subsampling (e.g. 1%, 10%) be more appropriate?

Accumulation curves were constructed using a technique similar to that suggested by the reviewer. We subsampled the number of metagenomes and tracked the number of genes for each species. This is a standard method adopted from pangenomics. We have rephrased this section to discuss “nearing” or “approaching” saturation instead of “reaching” saturation.

Line 231 – insert comma after 'saturation'

We have corrected this typo.

Line 233 – please revise: 'The non-redundant genes were decorated with a rich set of functional annotations'. 'decorated' seems an unusual choice.

We have revised this sentence. “The non-redundant genes were annotated with a rich set of functional descriptions.”

Line 234 – eliminate ‘both’

The change was made as suggested.

Line 247 – replace ‘interrogate’ with ‘investigate’

The change was made as suggested.

Line 246-247 – this sentence can be eliminated, similar sequence in Line264-266

We have removed the sentence from line 264-266 and rewritten the introduction to this paragraph, which now reads:

“To demonstrate the utility of VOGs, we retrieved 32 proteins of the orthologous family encoding vaginolysin, a *G. vaginalis* cholesterol-dependent cytolysin that is key to its pathogenicity as it forms pore in epithelial cells²⁹ (**Supplementary Table 6, Supplementary Fig. 9**).”

Line 248 – eliminate ‘Briefly’

Change made as suggested.

Line 261 – Files should be available

These files are now available on the provided link www.virgo.igs.umaryland.edu.

Line 271 – replace ‘a’ with ‘an’

Change made as suggested.

Line 271 – break sentence between ‘vaginolysin’ and ‘one’ Line 272 – eliminate ‘had’

Change made as suggested.

Line 276 - revise beginning of sentence: ‘As another example to use VOG for a large-scale data mining of protein family of interest,...’

We have revised this sentence as suggested. The sentence now reads:

“To provide another example of how to use the VOG database to mine genomic information, we searched VOG using the key phrase “cell surface-associated proteins” and “*L. iners*” and retrieved two protein families, one of which was recognized to have an LPXTG motif while the other harbored the motif YSIRK (**Supplementary Table 7**).”

*Line 308-317 – ‘L. iners-dominant communities...’ Are the authors referring to CST III only? If not, please describe how dominant was defined? This finding is not very surprising. Unlikely *L. crispatus*, *L. iners* seems to be found, more often than not, in microbiomes profiles with very diverse community composition. This would explain the classification into the HGC group and also the difference observed between LGC and HGC. Similar to *L. iners*, *G. vaginalis* can found as a dominant community (more than 10% of the metagenomic samples have 80% or more *G. vaginalis*) as well as part of a very diverse community.*

The reviewer is correct, we refer to CST III only (*L. iners*-dominant communities)

Line 329-33 – Is there a statistical analysis to support this?

This is one observation we included to simply highlight how HGC and LGC genes can be informative to study this specific tryptophan biosynthesis genes.

*Line 340 – Use *Mageeibacillus indolicus* instead of *BVAB3**

We have made this change throughout the manuscript and supplemental files.

Line 341-345 – Please revise sentence for clarity

We have revised this sentence as suggested. The sentence now reads:

These results demonstrate how gene richness-based annotations provide an added dimension to our understanding of the genetic basis of the biological processes that drive vaginal microbiomes.

Line 363-364 – ‘(TB3)’ should immediately follow ‘BV episode’

We have made this change.

Line 378 – There is no details regarding the 1507 metagenome samples. Please provide the accession numbers. Are these samples part of a separate manuscript? Please provide citation.

We have made these metagenomic data available and provided a link in the manuscript. A specific page on our institutional website has been dedicated to VIRGO and now serves this very large amount of the data.

Line 382 – make it possessive ‘species’

This has been corrected.

Line 394 – eliminate ‘basically’ Line 411 – revise sentence

We have made this change.

Line 414 – what well-established practices?

These methods are described in the cited references and in the sentences immediately following. “We next applied well-established practices from pangenomics⁵ in order to identify core and accessory non-redundant genes among our sample-specific species gene repertoires. Based on the clustering patterns of gene prevalence profiles, we were able to define groups of consistently present (core) and variably present (accessory) non-redundant genes. “

Line 438-441 – revise sentence

We have revised this sentence as suggested by the reviewer.

“Our analysis of *G. vaginalis* revealed more than four MG-subspecies, concordant with previously studies that had identified multiple clades within this species³². However, we found that the genome-based

paradigm largely under-represents the diversity of *G. vaginalis* gene content that we identified in metagenomes (**Supplementary Fig. 13e**).”

Line 445 – replace ‘interrogation’ with ‘investigation’

This change has been made.

Line 554 – replace ‘referenced’ with ‘reference’

We have corrected this typo.

Line 587-588 – This sentence seems dubious ‘416 urogenital bacterial strains and 321 bacterial species’. There are ~146 bacterial species, please clarify.

We have corrected this information. The 416 versus 321 were referring to the number of genomes versus unique genomes. The text correctly refers to the number of species as 153 (after including the addition of *Sneathia*, as suggested by the reviewer, and BVAB1.) “After removing duplicate genomes under the same strain names, genomes of 321 urogenital bacterial strains representing 153 bacterial species were included in the catalog.”

Line 596 – Eliminate “Briefly”

This change has been made.

Line 631 – Please explain “Sequencing libraries of A and B containing 6 bp..”

We have revised this sentence which now reads:

“Sequencing libraries were prepared using the TruSeq RNA sample prep kit (Illumina) following a modification of the manufacturer’s protocol: cDNA was purified between enzymatic reactions and library size selection was performed with AMPure XT beads (Beckman Coulter Genomics). Library sequencing was performed using the Illumina HiSeq 2500 platform (150 bp paired end mode, eight samples per lane).”

Methods – Please describe read length, etc... how many samples per lane, Q15 seems to low

This information has been added to the methods. MG and MT were sequenced on the HiSeq 2500 in 150 bp paired end mode. Eight samples were pooled in each lane (see comment above). The Q15 score refers to the average quality within the sliding window. We have revised the quality trimming sentence to more accurately reflect what we had done. “(3) stringent quality control using Trimmomatic v0.36⁴⁶, in which the Illumina adapters were excised, reads were trimmed using a 4bp sliding window with an average quality score threshold of Q15, and reads containing any ambiguous bases were removed. ”

Line 725-727 – BVAB1 is not listed in Suppl Table 1, please review this sentence for details and clarification. which database was used to annotate BVAB1 genes? Why there is a different pipeline for BVAB1?

The BVAB1 genome was not available until recently. However, because VIRGO was built from metagenomic data, it inherently contains the genetic contents of most vaginal bacteria. For this reason, we were able to identify the genes that belong to this species in VIRGO.

Line 755 – ‘Their accession numbers can be found in Supplementary Table 12.’ Suppl. Table 12 has Taxonomic distribution of pullulanase domain-containing proteins included in VIRGO.

The 1,507 metagenomes are now available at www.virgo.igs.umaryland.edu. The text was changed and now reads:

“A total of 1,507 vaginal metagenomes including 1,403 *in-house* from de-identified vaginal swab and lavage specimens and 76 publicly available, were mapped against VIRGO. These metagenomic datasets are available at www.virgo.igs.umaryland.edu.”

Line 1175 - Figure 3. How the genome size of BVAB1 was determined?

The BVAB1 genome was closed by our group and is described in the pending publication as cited.

Reference 11 is not cited. Remove randomly capitalized words.

This reference has been removed. Capitalization has been fixed throughout.

Legend Figure 6d. At the top bar, what are the colors blue, green and gray representing?

We have corrected the legend to more clearly describe these color annotations.

Title of Suppl Table 3 indicates ‘212 in-house’

This has been corrected to 211.

*Supp Table 4 (multiple tables and throughout the manuscript) – Use *Mageeibacillus indolicus* instead of BVAB3*

We have corrected this in Supplementary Tables 3,4 and 12.

Lachnospiraceae_bacterium (row 225), there are many members of this family, please specify. Don't italicize family name, Lachnospiraceae. Column C refers to genera, Lachnospiraceae is not a genus.

We follow the recommendation of the American Society of Microbiology guidelines which states that all taxonomic levels should be italicized. We have corrected the column heading to refer to “Genus/Family”.

We could not assign a genus or species to these 7 genes because the isolate genome sequence used to annotate them was only named “*Lachnospiraceae bacterium*”.

Reviewers' comments:

Reviewer #1 (Remarks to the Author):

General

The authors have addressed the two large issues of data availability and usability by providing a stable web portal where data can be accessed and downloaded, and an updated GitHub repo for the scripts and workflow. This is satisfactory for the support of the publication and usability of the database and tools for the research community. The authors have also put effort to update the database for missing reference genomes and incorrect taxonomy labels.

There are a few pieces of methodology not described, as well as some citations that could support the author's findings from the literature.

Text

P4: "Previous characterization of these communities using either 16S rRNA gene taxonomic profiling or assembly based metagenomic analyses has failed to resolve this diversity. Here we show that vaginal communities contain far more intraspecies diversity than originally expected. This observation challenges the notion that the vaginal microbiota dominated is by one species of *Lactobacillus*, comprised of a single strain, and could have major implications for the ecology of these otherwise low-diversity bacterial communities."

- Although most 16S amplicon studies are limited in ability to resolve intraspecies variation, I think it's unfair to say that the current dogma is so simplistic since there have been several whole genome and comparative genomics studies providing information on the diversity of the bacterial strains e.g.

van der Veer, Charlotte, et al. "Comparative genomics of human *Lactobacillus crispatus* isolates reveals genes for glycosylation and glycogen degradation: implications for in vivo dominance of the vaginal microbiota." *Microbiome* 7.1 (2019): 49.

Mendes-Soares, Helena, et al. "Comparative functional genomics of *Lactobacillus* spp. reveals possible mechanisms for specialization of vaginal lactobacilli to their environment." *Journal of bacteriology* 196.7 (2014): 1458-1470.

Schellenberg, John J., Mo H. Patterson, and Janet E. Hill. "Gardnerella vaginalis diversity and ecology in relation to vaginal symptoms." *Research in microbiology* 168.9-10 (2017): 837-844.

Ojala, Teija, et al. "Comparative genomics of *Lactobacillus crispatus* suggests novel mechanisms for the competitive exclusion of *Gardnerella vaginalis*." *BMC genomics* 15.1 (2014): 1070.

P15: Pullulanases have been noted in vaginal bacteria by other groups prior to this e.g.

Ojala, Teija, et al. "Comparative genomics of *Lactobacillus crispatus* suggests novel mechanisms for the competitive exclusion of *Gardnerella vaginalis*." *BMC genomics* 15.1 (2014): 1070.

Macklaim, Jean M., et al. "At the crossroads of vaginal health and disease, the genome sequence of *Lactobacillus iners* AB-1." *Proceedings of the National Academy of Sciences* 108.Supplement 1 (2011): 4688-4695.

"To retrieve pullulanase (pula) genes in VIRGO, we used conserved protein domain CDD54 annotation and keyword "pullulanase""

- Does this mean the representative proteins for this gene were extracted and used for sequence comparison? How was the matching done?

"For each of the seven species, a presence/absence matrix for the species' non-redundant genes was constructed"

- How was gene presence or absence determined?

P7: It would help readability to include the general characteristics (e.g. dominant composition) of the CTS reported. How does this distribution compare to other reported populations?

Fig 1/P7

- Why are samples from African women more highly mapped in the non-VIRGO databases?

- What processing/filtering/exclusion parameters were used for the external data before mapping to the comparative databases? e.g. was human contamination removed prior to mapping

L164-166

"Metagenomic assemblies contributed ~80% of the CDSs while the remaining ~20% originated from urogenital bacteria isolate genome sequences."

- Why were metagenomes favoured over whole genomes for CDS annotation considering that metagenomes could have misassemblies, chimeric contigs, etc. ?

- Later text seems to support the unreliability of metagenome assembly: "Unfortunately, intraspecies diversity is difficult to detect using typical assembly-based metagenomic analysis strategies, which are notoriously ill suited for resolving strains of the same species"

L 208

"This pales in comparison to the number of non-redundant genes included in VIRGO for *G. vaginalis*, which surpasses 25,000 genes. *G. vaginalis* is the only species analyzed which, as estimated by metagenome accumulation curves, did not approach saturation."

- This is an interesting observation, but it is not surprising considering recent literature exploring *G. vaginalis* phenotype and gene content, and proposals to include more species to the *Gardnerella* group. The authors should acknowledge some of this work that supports their observations:

Yeoman, Carl J., et al. "Comparative genomics of *Gardnerella vaginalis* strains reveals substantial differences in metabolic and virulence potential." *PloS one* 5.8 (2010): e12411.

Cornejo, Omar E., et al. "Focusing the diversity of *Gardnerella vaginalis* through the lens of ecotypes." *Evolutionary applications* 11.3 (2018): 312-324.

Ahmed, Azad, et al. "Comparative genomic analyses of 17 clinical isolates of *Gardnerella vaginalis* provide evidence of multiple genetically isolated clades consistent with subspeciation into genovars." *Journal of bacteriology* 194.15 (2012): 3922-3937.

Castro, Joana, Kimberly K. Jefferson, and Nuno Cerca. "Genetic Heterogeneity and Taxonomic Diversity among *Gardnerella* Species." *Trends in microbiology* (2019).

L 264-266

"Using the retrieved alignment, we identified 3 amino acid variants in an 11-amino acid sequences of domain 4 of vaginolysin. One of the three variants, an alanine-to-valine substitution that is divergent across *G. vaginalis* and not been reported previously."

- It's not clear to me which variant position had not been reported previously. The one highlighted in Supp Fig 9 is well-described here:

Gelber, Shari E., et al. "Functional and phylogenetic characterization of Vaginolysin, the human-specific cytolyisin from *Gardnerella vaginalis*." *Journal of bacteriology* 190.11 (2008): 3896-3903.

L 270-277

"To provide another example of how to use the VOG database to mine genomic information, we searched VOG using the key phrase "cell surface- associated proteins" and "L. iners" and retrieved two protein families, one of which was recognized to have an LPXTG motif while the other harbored the motif YSIRK (Supplementary Table 7). Interestingly, a previous study on staphylococcal proteins suggested that the motifs LPXTG and YSIRK were involved in different biological processes related to surface protein anchoring to cell wall envelope³⁰. The two retrieved protein families are specific to *L. iners* and provide relevant evidence for future experimental validation to understand its adherence."

This has been reported previously:

Macklaim, Jean M., et al. "At the crossroads of vaginal health and disease, the genome sequence of *Lactobacillus iners* AB-1." *Proceedings of the National Academy of Sciences* 108.Supplement 1 (2011): 4688-4695.

L 333

"*A. vaginae* exhibited the highest degree of intraspecies diversity, while *L. crispatus* has the highest within-metagenome intraspecies diversity among the major vaginal *Lactobacillus* spp."

- Please confirm and clarify in text that "intraspecies diversity" is measured within isolate genomes only, while "metagenome intraspecies diversity" is within metagenomic sequences only

Methods

"Procedures for DNA extraction and concentration qualification were previously described¹⁷"

- Could you please include the complete methodology in the text or supplemental information

Some of the bioinformatics tool choices are a little odd and out-of-date when there are currently better options available. For e.g.

- BMTagger has not been updated since 2014 nor peer-reviewed published
- IDBA-UD first published in 2009 and has not been updated in 6 years
- MetageneMark (v3.25) published in 2010
- Bowtie1 has generally been outfavoured by bowtie2

Please indicate which versions of the databases were used. e.g. SILVA PARC db, KEGG, CDD, Pfam etc.

For all software can you provide parameters used either in text or code provided as supplemental information.

E.g. MetaPhlan2, rarification (which tool? to what read depth?)

Reviewer #2 (Remarks to the Author):

I am glad to see that several points raised before were addressed by the authors, including the addition of very relevant bacterial taxa, data availability and usability.

Thanks for the information on how the genomes were retrieved. As the authors stated, VIRGO is a human vaginal non-redundant gene catalog, so it is my understanding that the ecosystem should be restricted to humans and type/subtype/keyword should include either 'vagina' or 'reproductive system' or 'urogenital system'. The title of the Suppl. Table 1 – worksheet #4 is "List of genomes isolated from vagina, reproductive or urinary system deposited in GenBank". With this description, the authors are stating that all genomes listed in this table were isolated from the vagina, reproductive or urinary systems. This statement is not correct. This concern was raised previously using as an example the Salmonella genomes. Looking closely, the majority of the BioSample records for the Salmonella genomes listed in this table indicate the Isolation Source as 'environmental samples' or 'eggs'. One of the listed genomes was isolated from "Chicken Ovary". I would like to add

another example, the taxon *Ornithobacterium rhinotracheale* DSM 15997 (# 867902), listed at row 214. The BioSample (SAMN02261366) record indicates the host is 'Turkey', Body Sample Site is 'Airways' and Body Sample Subtype is 'Respiratory tract'. There is no indication that this particular taxon was ever isolated from the vagina, reproductive or urinary systems. Looking further, there are three additional genomes available for *Ornithobacterium rhinotracheale* and all three BioSample records similarly indicate that the host is avian/poultry and body site is related to the respiratory tract. To my knowledge this taxon has never been described in humans or any reproductive/ urinary systems. Based on the examples described above, it is not correct to list these genomes as "isolated from vagina, reproductive or urinary system" and it is not clear how these genomes were included using the described selection criteria.

Line 135/Suppl. Table 1 – worksheet #4 (Suppl. Table 1) entitled 'List of genomes isolated from vagina, reproductive or urinary system deposited in GenBank' contains genomes isolated from 'environmental swabs' as indicated for instance in ncbi taxon ids 910413 and 910414. Please clarify or make appropriate corrections. GenBank Assembly ID of the genomes used to build the database should be provided. The list of genomes was retrieved from genBank on Nov 9th, 2015. They were retrieved based on the following criteria of the metadata provided on GenBank report: 1) The ecosystem is "host-associated" and the ecosystem subtype is "vagina"; 2) If no ecosystem subtype was specified, ecosystem type is listed as "reproductive system"; 3) If neither ecosystem or ecosystem subtypes was specified, the habitat contain the key word "vagina". We used these criteria to retrieve genomes because the genBank metadata is not always in a consistent format and we tried to include a comprehensive, unbiased set of genomes associated with urogenital system. The *Salmonella* genomes were not removed from our list because they met the selection criteria. VIRGO results are not affected by the inclusion of these *Salmonella* genomes because, unless the species is present reads will not be mapped onto them. In our analysis of 1,507 metagenomes no *Salmonella* genes were identified.

Reviewer #1

General

The authors have addressed the two large issues of data availability and usability by providing a stable web portal where data can be accessed and downloaded, and an updated GitHub repo for the scripts and workflow. This is satisfactory for the support of the publication and usability of the database and tools for the research community. The authors have also put effort to update the database for missing reference genomes and incorrect taxonomy labels.

There are a few pieces of methodology not described, as well as some citations that could support the author's findings from the literature.

Text

P4: "Previous characterization of these communities using either 16S rRNA gene taxonomic profiling or assembly based metagenomic analyses has failed to resolve this diversity. Here we show that vaginal communities contain far more intraspecies diversity than originally expected. This observation challenges the notion that the vaginal microbiota dominated is by one species of *Lactobacillus*, comprised of a single strain, and could have major implications for the ecology of these otherwise low-diversity bacterial communities."

- Although most 16S amplicon studies are limited in ability to resolve intraspecies variation, I think it's unfair to say that the current dogma is so simplistic since there have been several whole genome and comparative genomics studies providing information on the diversity of the bacterial strains e.g.

van der Veer, Charlotte, et al. "Comparative genomics of human *Lactobacillus crispatus* isolates reveals genes for glycosylation and glycogen degradation: implications for in vivo dominance of the vaginal microbiota." *Microbiome* 7.1 (2019): 49.

Mendes-Soares, Helena, et al. "Comparative functional genomics of *Lactobacillus* spp. reveals possible mechanisms for specialization of vaginal lactobacilli to their environment." *Journal of bacteriology* 196.7 (2014): 1458-1470.

Schellenberg, John J., Mo H. Patterson, and Janet E. Hill. "Gardnerella vaginalis diversity and ecology in relation to vaginal symptoms." *Research in microbiology* 168.9-10 (2017): 837-844.

Ojala, Teija, et al. "Comparative genomics of *Lactobacillus crispatus* suggests novel mechanisms for the competitive exclusion of *Gardnerella vaginalis*." *BMC genomics* 15.1 (2014): 1070.

P15: Pullulanses have been noted in vaginal bacteria by other groups prior to this e.g.

Ojala, Teija, et al. "Comparative genomics of *Lactobacillus crispatus* suggests novel mechanisms for the competitive exclusion of *Gardnerella vaginalis*." *BMC genomics* 15.1 (2014): 1070.

Macklaim, Jean M., et al. "At the crossroads of vaginal health and disease, the genome sequence of *Lactobacillus iners* AB-1." *Proceedings of the National Academy of Sciences* 108.Supplement 1 (2011): 4688-4695.

We thank the reviewer for these suggestions. However, these studies have demonstrated that there is diversity within the key vaginal taxa (e.g. *Lactobacillus crispatus*, *G. vaginalis*) but have not addressed whether this diversity exists within a women's community, which we specifically addressed, it is well acknowledged that genomic diversity exists between isolates of the same species and we acknowledge this point. As the reviewer points out, the use of 16S rRNA gene sequencing, and we argue that of assembly-based metagenomics analysis, largely precludes the identification of multiple strains of the same species within a sample.

The main results of our study is that intraspecies diversity exists within a vaginal microbial community. This is reflected in the manuscript title, and the header of the section the reviewer refers to. The suggested citations do not support within community intra-species genomic diversity, some of them are already cited or were included in the first submission of the paper, but were removed due to the 70-citation limit imposed by the journal.

"To retrieve pullulanase (pula) genes in VIRGO, we used conserved protein domain CDD54 annotation and keyword "pullulanase"'"

- Does this mean the representative proteins for this gene were extracted and used for sequence comparison? How was the matching done?

The CDD annotation is part of the comprehensive functional protein annotation that is included in the VIRGO database. The steps described above simply involve pulling out genes from VIRGO based on the keyword "pullulanase" and filtering by the targeted CDD protein domain. These sequences are from the VIRGO database and were used in this comparison.

"For each of the seven species, a presence/absence matrix for the species' non-redundant genes was constructed"

- How was gene presence or absence determined?

Gene presence was determined by the mapping of at least three metagenomic reads to that gene. Our analysis showed that using more gene did not change the outcome, while using less did. We have added this detail to the methods section which now reads on line 663:

"For each of the seven species, a presence/absence matrix for the species' non-redundant genes (at least 3 reads mapped onto the gene) was constructed..."

P7: It would help readability to include the general characteristics (e.g. dominant composition) of the CTS reported. How does this distribution compare to other reported populations?

We have added a short description of each CST on line 134 in the results section, which now reads:

*"The taxonomic profiles of 264 metagenomes were further shown to encompass the five previously reported vaginal community state types (CSTs)¹⁷, CST I (*L. crispatus*-dominated), II (*L. gasseri*-dominated), III (*L. iners*-dominated), IV (array of strict and facultative anaerobes), and V (*L. jensenii*-dominated) with frequencies in this set of metagenomes of 18.9%, 3.8%, 20.5%, 48.5%, and 8.3%, respectively (**Supplementary Fig. 2.***

Supplementary Table 2)."

Each CST is well-represented, which supports the comprehensiveness of VIRGO, including CST II and CST V, two CSTs that are often present at low frequencies in most population.

Fig 1/P7

- Why are samples from African women more highly mapped in the non-VIRGO databases?

- What processing/filtering/exclusion parameters were used for the external data before mapping to the comparative databases? e.g. was human contamination removed prior to mapping

We feel that we cannot speculate on the potential reasons for the higher mapping of African women samples to the non-VIRGO databases, considering the low number of samples used in this analysis. That said, it is possibly likely due to the lower number of reads per samples, but again, this is speculative at this time.

The datasets used to test the comprehensiveness of VIRGO were preprocessed in the same manner as those used to build VIRGO. We added this information in the method section on line 639, which now reads: "The sequences reads were first pre-processed as described above for VIRGO construction and mapped to the VIRGO contigs using bowtie...."

L164-166

"Metagenomic assemblies contributed ~80% of the CDSs while the remaining ~20% originated from urogenital bacteria isolate genome sequences."

- Why were metagenomes favoured over whole genomes for CDS annotation considering that metagenomes could have misassemblies, chimeric contigs, etc. ?

- Later text seems to support the unreliability of metagenome assembly: "Unfortunately, intraspecies diversity is difficult to detect using typical assembly-based metagenomic analysis strategies, which are notoriously ill suited for resolving strains of the same species"

Metagenomes were not favored in the construction of the VIRGO. We included all available isolate genomes at the time when the database was built, which provided 1.1 million CDSs whereas the metagenomic data contributed 4.4 million CDSs. The metagenomic data is needed to capture diversity which is uncultivable or under-represented in the set of available isolate genomes.

It is true that assembled metagenomic data often fails to capture multiple strains of the same species within an individual sample. However, this diversity is provided to VIRGO by the inclusion of multiple metagenomes. Thus, while an individual assembly might miss a strain which is at low relative abundance, the CDSs from that strain could be provided by its appearance in a different metagenome where it is present at a higher relative abundance, or the presence of contig of lower coverage.

L 208

"This pales in comparison to the number of non-redundant genes included in VIRGO for *G. vaginalis*, which surpasses 25,000 genes. *G. vaginalis* is the only species analyzed which, as estimated by metagenome accumulation curves, did not approach saturation."

- This is an interesting observation, but it is not surprising considering recent literature exploring *G. vaginalis* phenotype and gene content, and proposals to include more species to the Gardnerella group. The authors should acknowledge some of this work that supports their observations:

Yeoman, Carl J., et al. "Comparative genomics of *Gardnerella vaginalis* strains reveals substantial differences in metabolic and virulence potential." *PloS one* 5.8 (2010): e12411.

Cornejo, Omar E., et al. "Focusing the diversity of *Gardnerella vaginalis* through the lens of ecotypes." *Evolutionary applications* 11.3 (2018): 312-324.

Ahmed, Azad, et al. "Comparative genomic analyses of 17 clinical isolates of *Gardnerella vaginalis* provide evidence of multiple genetically isolated clades consistent with subspeciation into genovars." *Journal of bacteriology* 194.15 (2012): 3922-3937.

Castro, Joana, Kimberly K. Jefferson, and Nuno Cerca. "Genetic Heterogeneity and Taxonomic Diversity among *Gardnerella* Species." *Trends in microbiology* (2019).

We agree that the finding of diversity within *G. vaginalis* is not novel when considered across samples. We would like to cite the papers listed but cannot due to the strict citation limit. However, the paper highlights a novel finding where multiple *G. vaginalis* subtypes are observed within an individual community. This is described later in the text on line 335 which reads: "Among these species, *G. vaginalis* and *A. vaginae* exhibited the highest within-metagenome intraspecies diversity,..." and the point argued by the reviewer is highlighted further on line 370:

"Our analysis of *G. vaginalis* revealed more than four MG-subspecies, concordant with previously studies that had identified multiple clades within this species³²"

L 264-266

"Using the retrieved alignment, we identified 3 amino acid variants in an 11-amino acid sequences of domain 4 of vaginolysin. One of the three variants, an alanine-to-valine substitution that is divergent across *G. vaginalis* and not been reported previously."

- It's not clear to me which variant position had not been reported previously. The one highlighted in Supp Fig 9 is well-described here:

Gelber, Shari E., et al. "Functional and phylogenetic characterization of Vaginolysin, the human-specific cytolysin from *Gardnerella vaginalis*." *Journal of bacteriology* 190.11 (2008): 3896-3903.

The specific variant in question was identified in the paper mentioned by the reviewer as being a *conserved* difference between *G. vaginalis* and other taxa, a paper we have cited in the manuscript. Here we have shown that this AA substitution is divergent within *Gardnerella*. Some strains have the Alanine residue, while other have the Valine residue at that position.

L 270-277

"To provide another example of how to use the VOG database to mine genomic information, we searched VOG using the key phrase "cell surface- associated proteins" and "L. iners" and retrieved two protein families, one of which was recognized to have an LPXTG motif while the other harbored the motif YSIRK (Supplementary Table 7). Interestingly, a previous study on staphylococcal proteins suggested that the motifs LPXTG and YSIRK were involved in different biological processes related to surface protein anchoring to cell wall envelope³⁰. The two retrieved protein families are specific to *L. iners* and provide relevant evidence for future experimental validation to understand its adherence." This has been reported previously:

Macklaim, Jean M., et al. "At the crossroads of vaginal health and disease, the genome sequence of *Lactobacillus iners* AB-1." *Proceedings of the National Academy of Sciences* 108.Supplement 1 (2011): 4688-4695.

We apologize for the omission and we have now cited this manuscript in the text on line 277, which now reads: "The two retrieved protein families are specific to *L. iners* and provide relevant evidence for future experimental validation to understand its adherence as previously reported³⁰."

L 333

"*A. vaginae* exhibited the highest degree of intraspecies diversity, while *L. crispatus* has the highest within-metagenome intraspecies diversity among the major vaginal *Lactobacillus* spp."

- Please confirm and clarify in text that "intraspecies diversity" is measured within isolate genomes only, while "metagenome intraspecies diversity" is within metagenomic sequences only

The sentence was modified for clarity and now reads on line 335: "Among these species, *G. vaginalis* and *A. vaginae* exhibited the highest within-metagenome intraspecies diversity, while that of *L. crispatus* was the highest among the major vaginal *Lactobacillus* spp."

Methods

"Procedures for DNA extraction and concentration qualification were previously described¹⁷"

- Could you please include the complete methodology in the text or supplemental information

The methodologies have been extensively previously described and exact details on the methods for DNA extraction and qualification can be found in the cited papers. It would be redundant to repeat it here.

Some of the bioinformatics tool choices are a little odd and out-of-date when there are currently better options available. For e.g.

- *BMTagger has not been updated since 2014 nor peer-reviewed published*
- IDBA-UD first published in 2009 and has not been updated in 6 years
- *MetageneMark (v3.25) published in 2010*
- *Bowtie1 has generally been outfavoured by bowtie2*

We believe that our choices of bioinformatic tools were appropriate to achieve the study goals:

BMTagger is the tool provided by the HMP, and specifically designed for the removal of human DNA contaminants for public deposition of data. BMTagger performs extremely well at removing human DNA from metagenomic sequencing dataset. In our experience, BMTagger is more sensitive than other alternative, which were not design with that specific function in mind (i.e., Bowtie or BWA).

Bowtie 1 and 2 are used for different purposes: Bowtie1 can be used for reads mapping at nucleotide level to increase the match sensitivity while Bowtie 2 can be used for “gapped” read mapping to increase the chance of searching for distantly related sequences to increase “recall” rate. For the purpose of this paper we aimed for close-matching nucleotide sequences, thus bowtie 1 was the appropriate tool to use.

IDBA-UD was commonly used to assemble metagenomic data at the time the database was built. Results are unlikely to change dramatically with the use of other assembly algorithms (e.g. metaspades).

Please indicate which versions of the databases were used. e.g. SILVA PARC db, KEGG, CDD, Pfam etc.

For all software can you provide parameters used either in text or code provided as supplemental information. E.g. MetaPhlan2, rarification (which tool? to what read depth?)

The version numbers were added to the text and are now listed on (also on Nature Communications Reporting Summary form):

line 609:

“Functional annotations based on the standard procedure for each of 17 functional databases, including: cluster of orthologous groups (COG (v1)⁵¹, eggNOG (v4.5)⁵², KEGG (FTP Release 2013-03-18)⁵³), conserved protein domain (CDD (v3.14)⁵⁴, Pfam (v30.0)⁵⁵, ProDom (v20.119)⁵⁶, PROSITE (V20.119)⁵⁷, TIGRFAM (v15.0)⁵⁸, InterPro (v60.0)⁵⁹), domain architectures (CATH-Gene3D (v4.1)⁶⁰, SMART (v7.1)⁶¹), intrinsic protein disorder (MobiDB (v2.0)⁶²), high-quality manual annotation (HAMAP (v201605.11)⁶³), protein superfamily (PIRSF (v3.01)⁶⁴), a compendium of protein fingerprints (PRINTS⁶⁵), and gene product attributes (Gene Ontology , JCVI SOP²⁷).”

then on line 561: “(2) *in silico* microbial rRNA sequence reads depletion by aligning all reads using Bowtie (v1)⁴⁵ against the SILVA PARC ribosomal-subunit sequence database (v132)⁹ to eliminate mis-assemblies of these repeated regions.

Reviewer #2

I am glad to see that several points raised before were addressed by the authors, including the addition of very relevant bacterial taxa, data availability and usability. I have a follow up question and the details can be found in the attached document. PLEASE SEE ATTACHMENT.

Thanks for the information on how the genomes were retrieved. As the authors stated, VIRGO is a human vaginal non-redundant gene catalog, so it is my understanding that the ecosystem should be restricted to humans and type/subtype/keyword should include either 'vagina' or 'reproductive system' or 'urogenital system'. The title of the Suppl. Table 1 – worksheet #4 is "List of genomes isolated from vagina, reproductive or urinary system deposited in GenBank". With this description, the authors are stating that all genomes listed in this table were isolated from the vagina, reproductive or urinary systems. This statement is not correct.

This concern was raised previously using as an example the Salmonella genomes. Looking closely, the majority of the BioSample records for the Salmonella genomes listed in this table indicate the Isolation Source as 'environmental samples' or 'eggs'. One of the listed genomes was isolated from "Chicken Ovary".

I would like to add another example, the taxon Ornithobacterium rhinotracheale DSM 15997 (# 867902), listed at row 214. The BioSample (SAMN02261366) record indicates the host is 'Turkey', Body Sample Site is 'Airways' and Body Sample Subtype is 'Respiratory tract'. There is no indication that this particular taxon was ever isolated from the vagina, reproductive or urinary systems. Looking further, there are three additional genomes available for Ornithobacterium rhinotracheale and all three BioSample records similarly indicate that the host is avian/poultry and body site is related to the respiratory tract. To my knowledge this taxon has never been described in humans or any reproductive/ urinary systems.

Based on the examples described above, it is not correct to list these genomes as "isolated from vagina, reproductive or urinary system" and it is not clear how these genomes were included using the described selection criteria.

Previous response from the authors:

Line 135/Suppl. Table 1 – worksheet #4 (Suppl. Table 1) entitled 'List of genomes isolated from vagina, reproductive or urinary system deposited in GenBank' contains genomes isolated from 'environmental swabs' as indicated for instance in ncbi taxon ids 910413 and 910414. Please clarify or make appropriate corrections. GenBank Assembly ID of the genomes used to build the database should be provided.

The list of genomes was retrieved from genBank on Nov 9th, 2015. They were retrieved based on the following criteria of the metadata provided on GenBank report: 1) The ecosystem is "host-associated" and the ecosystem subtype is "vagina"; 2) If no ecosystem subtype was specified, ecosystem type is listed as "reproductive system"; 3) If neither ecosystem or ecosystem subtypes was specified, the habitat contain the key word "vagina". We used these criteria to retrieve genomes because the genBank metadata is not always in a consistent format and we tried to include a comprehensive, unbiased set of genomes associated with urogenital system. The Salmonella genomes were not removed from our list because they met the selection criteria. VIRGO results are not affected by the inclusion of these Salmonella genomes because, unless the species is present reads will not be mapped onto them. In our analysis of 1,507 metagenomes no Salmonella genes were identified.

Response from the authors:

We thank the reviewer for the thorough evaluation. While, the annotation at time of retrieval (2015) was associated with the keyword "reproductive system", the NCBI database has been updated since and these genomes do not show this annotation anymore, while the database we downloaded shows the annotation. That

said, none of the 1,507 metagenomes analyzed in this study mapped to these organisms, thus we have removed them from the database and supplementary table 1 and 4 wer updated to reflect this change.

As a consequence, the number of genomes used in the analysis have been also updated in the text on line 188 and line 503.